# Revisiting Multi-Permutation Equivariance through the Lens of Irreducible Representations

**Yonatan Sverdlov[1,*], Ido Springer[1,*] & Nadav Dym[1,2]**
[1]Faculty of Mathematics
[2]Faculty of Computer Science
Technion – Israel Institute of Technology
`nadavdym@technion.ac.il`
`{yonatans,ido.springer}@campus.technion.ac.il`

## Abstract

This paper explores the characterization of equivariant linear layers for representations of permutations and related groups. Unlike traditional approaches, which address these problems using parameter-sharing, we consider an alternative methodology based on irreducible representations and Schur's lemma. Using this methodology, we obtain an alternative derivation for existing models like DeepSets, 2-IGN graph equivariant networks, and Deep Weight Space (DWS) networks. The derivation for DWS networks is significantly simpler than that of previous results.

Next, we extend our approach to unaligned symmetric sets, where equivariance to the wreath product of groups is required. Previous works have addressed this problem in a rather restrictive setting, in which almost all wreath equivariant layers are Siamese. In contrast, we give a full characterization of layers in this case and show that there is a vast number of additional non-Siamese layers in some settings. We also show empirically that these additional non-Siamese layers can improve performance in tasks like graph anomaly detection, weight space alignment, and learning Wasserstein distances. Our code is available at GitHub.

## 1 Introduction

Learning with symmetries has recently attracted great attention in machine learning. In this learning setting, a group acts on an input space, and the hypothesis mappings are restricted to be equivariant with respect to the group action. Motivated by the structure of fully connected neural networks, group equivariant models are often defined by a composition of parametric linear equivariant functions and non-parametric non-linear activations functions (Cohen & Welling, 2016). Thus, characterizing all equivariant layers for a given group action is fundamental for understanding and designing equivariant deep neural networks. Indeed, this question has attracted a considerable amount of attention in various scenarios (e.g., Finzi et al. (2021); Kondor & Trivedi (2018); Cohen et al. (2019); Pearce-Crump (2023a;b)).

For permutation groups, the standard strategy for characterizing equivariant layers is through studying parameter-sharing (Ravanbakhsh et al., 2017). Perhaps the first result in this direction was given by Zaheer et al. (2017), who showed that only two parameters are required to characterize all permutation equivariant layers on a set of scalars. Another famous example is Maron et al. (2018), who described the specific parameter-sharing scheme for graph equivariant networks and other tensor representations of the symmetric group. Recently, parameter-sharing has been applied in Navon et al. (2023); Zhou et al. (2024b) to characterize all equivariant layers of weight space neural functionals. These neural networks operate on weights of another neural network, a problem with a complex multi-permutation equivariant structure.

In this paper, we revisit equivariant linear layer characterization for permutation groups from the perspective of irreducible representations. Given a group $\mathcal{G}$ acting on a linear space $\mathcal{V}$, we explore

---
*Equal contribution

linear equivariant mappings $f : \mathcal{V} \to \mathcal{V}$ by decomposing $\mathcal{V}$ to its minimal invariant components, which are irreducible representations of $\mathcal{G}$. Once this decomposition is performed, a basic result in representation theory called Schur's lemma automatically provides us with a full characterization of all equivariant mappings.

As a first step, we use our method to get an alternative derivation for known equivariant layer characterization at the basis of the DeepSets (Zaheer et al., 2017), 2-IGN (Maron et al., 2018), and DWSNets (Navon et al., 2023) architectures. Notably, for the Deep Weight Space problem, our derivation is significantly simpler than the cumbersome derivations previously obtained by parameter sharing (Navon et al., 2023; Zhou et al., 2024b).

Next, we consider the problem of sets of *unaligned* symmetric elements, where equivariance to the joint action of a group $\mathcal{G}$ and the permutation group $S_k$ is required (also known as the wreath product $\mathcal{G} \wr S_k$). This setting arises naturally when considering alignment, distance prediction, or anomaly detection of sets of unaligned symmetric elements. It was studied by Wang et al. (2020) under the assumption that $\mathcal{G}$ acts transitively, in which case Siamese Networks capture an overwhelming majority of the equivariant layers. Indeed, in practice, Siamese networks are often employed (Navon et al., 2024; Chen & Wang, 2024). In contrast, we give a full characterization of the equivariant layers in the general setting and show that in some settings (e.g., alignment of weights spaces), there can be a large number of non-Siamese equivariant layers. Empirically, we show that these additional layers improve performance on a synthetic graph anomaly detection task and the deep weight space alignment task discussed in Navon et al. (2024). In summary, our contributions are the following:

• We give an alternative irreducible-based derivation for the DeepSets (Zaheer et al., 2017), 2-IGN (Maron et al., 2018), and DWSNets (Navon et al., 2023) architectures. The DWSnets derivation is significantly simpler than previous approaches.

• We provide a characterization of all equivariant functions on sets of unaligned symmetric elements.

• We empirically validate the importance of our equivariant layers for sets of unaligned symmetric elements by showing its superiority on a synthetic graph anomaly detection task, Wasserstein set distance computation, and deep weight space alignment task.

## 2 RELATED WORK

**Learning Sets of Symmetric Elements.** When dealing with multiple occurrences of symmetric objects as a set, additional symmetries arise as the set elements can be permuted. In this case, the layers are required to be equivariant under the permutation of elements in addition to the original element group action. Maron et al. (2020) suggested a new layer called Deep Sets for Symmetric Elements (DSS) that operates on a set of symmetric objects. They showed that incorporating DSS layers is strictly more expressive than using Siamese networks. The DSS framework was also used in sub-graph aggregation networks (Bevilacqua et al., 2021), where the network should be invariant to the order of the sub-graphs. In the main setting of Maron et al. (2020), the same group element $g \in G$ acts uniformly on all set objects. However, the more challenging setting is where we allow different elements $g_i \in G$ to act on the objects. This corresponds to the action of the restricted wreath product $G \wr S_k$ on the set. Maron et al. (2020) deal with this setting as well, but only when $G$ is a subgroup of $S_n$ and the action of $G$ is transitive. Later, Wang et al. (2020) characterized all linear mappings for general wreath products, however they also assumed that the group actions are transitive.

**Equivariant Characterization Via Irreducibles.** Using irreducible representations to characterize equivariant layers is a popular approach for rotation-equivariant networks (Thomas et al., 2018; Anderson et al., 2019; Dym & Maron). For permutation groups, the primary work which considered this approach, to the best of our knowledge, is Henning Thiede et al. (2020). They suggest using the known characterization of permutation irreducible representations with Young diagrams (Fulton & Harris, 2013) to characterize permutation-equivariant layers and explain how to reconstruct the results from DeepSets and 2-IGN. Our approach for these results is similar but gives a more explicit derivation of the DeepSet and 2-IGN layers. More importantly, our analysis covers deep weight spaces and sets of unaligned symmetric elements, not discussed in this paper.

## 3 PRELIMINARIES

Let $\mathcal{V}$ be a finite-dimensional vector space over the field $\mathbb{F} = \mathbb{C}$ or $\mathbb{R}$. Let $\mathcal{G}$ be a finite group acting linearly on $\mathcal{V}$. We will sometimes say that $\mathcal{V}$ is a representation of $\mathcal{G}$. We will say $\mathcal{V}$ is a trivial representation of $\mathcal{G}$ if the action of $\mathcal{G}$ on $\mathcal{V}$ is the trivial action $gv = v$ for all $g \in \mathcal{G}$ and $v \in \mathcal{V}$.

A subspace $\mathcal{U} \subseteq \mathcal{V}$ is invariant to $\mathcal{G}$ if $g \cdot u \in \mathcal{U}$ for all $g \in G$ and $u \in \mathcal{U}$. We say an invariant subspace $\mathcal{U}$ is irreducible if it does not strictly contain an invariant subspace except the zero space. By Maschke's theorem, each finite-dimensional space $\mathcal{V}$ can be decomposed into a direct sum of irreducible invariant spaces (Fulton & Harris, 2013).

A mapping $T : \mathcal{V} \to \mathcal{V}$ is equivariant if $T(g \cdot v) = g \cdot T(v)$ for every $v \in V, g \in \mathcal{G}$. Two representations $\mathcal{V}, \mathcal{U}$ of $\mathcal{G}$ are isomorphic if there is an equivariant linear bijection $L_{\mathcal{V},\mathcal{U}} : \mathcal{V} \to \mathcal{U}$, which we notate by $\mathcal{V} \cong \mathcal{U}$.

The following lemma, named Schur's lemma, is a key component of our work. Schur's lemma describes equivariant mappings between irreducible representations:

**Schur's lemma.** Let $\mathcal{V}, \mathcal{W}$ be finite-dimensional irreducible representations of $\mathcal{G}$ over $\mathbb{F}$, and let $T : \mathcal{V} \to \mathcal{W}$ be a $\mathcal{G}$-equivariant linear map [*]. Then:

- Either $T$ is an isomorphism or $T = 0$.
- If $L : \mathcal{V} \to \mathcal{W}$ is an isomorphism, and $\mathbb{F} = \mathbb{C}$, then $T = \lambda L$ for $\lambda \in \mathbb{C}$.

In this paper, we will only consider representations over $\mathbb{R}$, which are the common setting in applications. In all cases, we will discuss, our real irreducible representations are *absolutely irreducible*. This means that their natural extension to complex vector spaces is irreducible. In this case, if $L : \mathcal{V} \to \mathcal{W}$ is an isomorphism, $\mathbb{F} = \mathbb{R}$, and $\mathcal{V}$ is *absolutely irreducible*, then $T = \lambda L$ for $\lambda \in \mathbb{R}$ (Boardman). In Appendix B we elaborate more on this topic. We note that when $V$ is not absolutely irreducible, the space of isomorphisms from $\mathcal{V}$ to $\mathcal{W}$ is either 2 or 4 dimensional (Poonen, 2016). In this setting, using an automatic computational method to find all 2-4 equivariant layers may be beneficial (Finzi et al., 2021).

**Layer characterization using Schur's Lemma** Equivariant layers from a representation $\mathcal{V}$ to itself can be characterized using Schur's lemma via two steps. The first step is decomposition, where we identify the decomposition of $\mathcal{V}$ into irreducible representations $\mathcal{V} = \oplus_{i=1}^{s} \mathcal{V}_i$. Any $x \in \mathcal{V}$ can be written uniquely as a sum $x = \sum_{i=1}^{s} x_i$ where each $x_i$ is in $\mathcal{V}_i$. The decomposition step also requires an algorithm to compute this decomposition. Next, we will need to identify which of the space pairs $\mathcal{V}_i$ and $\mathcal{V}_j$ are isomorphic, and if they are, specify an isomorphism $L_{ij}$. Finally, since we are working over the reals, we will need to verify that all irreducible representations are absolutely irreducible.

Once the decomposition step is carried out, the second step uses Schur's lemma. Equivariant mappings $T : \mathcal{V} \to \mathcal{V}$ can be written as $T = \sum_{i,j=1}^{s} T_{ij}$, where $T_{ij} : \mathcal{V}_i \to \mathcal{V}_j$ are equivariant. By Schur's lemma if $\mathcal{V}_i$ and $\mathcal{V}_j$ are isomorphic then $T_{ij} = \lambda_{ij} L_{ij}$ for some $\lambda_{ij} \in \mathbb{R}$, and otherwise $T_{ij} = 0$. All in all, we obtain that $T$ is equivariant if and only if for some choice of the scalar $\lambda_{ij}$ we have

$$T(x_1 + \ldots + x_s) = \sum_{(i,j), \mathcal{V}_i \cong \mathcal{V}_j} \lambda_{ij} L_{ij}(x_i). \tag{1}$$

The number of parameters in the expression above depends on the isomorphism relations between the irreducibles. We can assume without loss of generality that the first $1 \leq t \leq s$ irreducible spaces are not isomorphic, and by identifying isomorphic irreducibles, we can rewrite the decomposition above as $\mathcal{V} = \oplus_{j=1}^{t} \mathcal{V}_j^{\oplus \alpha_j}$ (Where $\mathcal{V}_j^{\oplus \alpha_j}$ is defined to be the direct sum of $\alpha_j$ copies of $\mathcal{V}_j$) where $\alpha_1 + \ldots + \alpha_t = s$. The number of parameters $\lambda_{ij}$ in the decomposition is then $\sum_{i=1}^{t} \alpha_i^2$. This process can easily be generalized to compute mappings between representations $\mathcal{V}$ and $\mathcal{W}$ by decomposing both spaces into irreducibles.

We note that the cornerstones of this methodology: decomposition into irreducibles and Schur's Lemma, are applicable for all finite dimensional representations of finite groups (and also for compact

---

[*]The equivariant map is often referenced as a $G$-module.

infinite groups like $SO(d)$). The main challenge in this approach is characterizing and computing the decomposition into irreducibles. This needs to be done on a case to case basis. Much of the remainder of the paper will be devoted to computing these decompositions for important equivariant learning scenarios.

## 4 COMPUTING LINEAR EQUIVARIANT LAYERS

We now demonstrate how to derive the results in Zaheer et al. (2017); Maron et al. (2018); Navon et al. (2023) via decomposition into irreducibles.

### 4.1 DEEP SETS

We begin with the simple case of the action of the permutation group $S_n$ on $\mathcal{V} = \mathbb{R}^n$ by permutation of elements (we assume $n \geq 2$). Here, there are two non-trivial invariant spaces:

$$\mathbf{S} = \{\alpha \cdot 1_n | \alpha \in \mathbb{R}\}, \quad \mathbf{V}(n) = \{x \in \mathbb{R}^n : \sum_{i=1}^{n} x_i = 0\}. \tag{2}$$

As the first space is one-dimensional, it's irreducible. By Hinton et al. (2006), the second one is also irreducible. These two spaces are not isomorphic since the action of $S_n$ on $\mathbf{S}$ is trivial, but the action of $S_n$ on $\mathbf{V}(n)$ is not (also they do not have the same dimension when $n \geq 3$).

The decomposition of $x \in \mathbb{R}^n$ to a sum of elements from the irreducible spaces can be computed as

$$x = \bar{x}1_n + (x - \bar{x}1_n)$$

where $\bar{x}$ is the average of $x$, and $1_n$ is the all one vector. By Equation (1), Schur's lemma, and the fact that all irreducibles of the permutation group are absolutely irreducible (see Appendix B), the linear equivariant mappings $T : \mathcal{V} \to \mathcal{V}$ are characterized by two parameters $a, b \in \mathbb{R}$, that is:

$$Tx = a\bar{x}1_n + b(x - \bar{x}1_n).$$

This is exactly the result described in DeepSets (Zaheer et al., 2017).

### 4.2 EQUIVARIANT GRAPH LAYERS

We next consider the setting where $\mathcal{V} = \mathbb{R}^{n \times n}, \mathcal{G} = S_n$ and the group action is defined by $(\tau \cdot X)_{ij} = X_{\tau^{-1}(i), \tau^{-1}(j)}$. This setting is natural for graph neural networks, as two adjacency matrices $A, B \in \mathbb{R}^{n \times n}$ represent isomorphic graphs if and only if $A = \tau \cdot B$ for some permutation $\tau$. This setting was considered in Maron et al. (2018) using a parameter sharing scheme, and we show how to obtain similar results using irreducibles (note that Maron et al. (2018) also discussed general mappings between $k$-order and $\ell$-order tensors. In contrast, we only discuss the $k = \ell = 2$ case).

We claim that, when $n \geq 4$, the space $\mathbb{R}^{n \times n}$ can be written as a sum of seven irreducible permutation invariant sub-spaces. The behavior for $n < 4$ is discussed in Appendix C.3.

The first two spaces are one-dimensional representations on which the action of $S_n$ is trivial: diagonal matrices with identical diagonal entries and matrices with zero diagonal and identical off-diagonal entries:

$$\mathcal{V}_0 = \{a \cdot I_n | a \in \mathcal{R}\}, \quad \mathcal{V}_1 = \{a \cdot (1_{n \times n} - I_n) | a \in \mathcal{R}\}. \tag{3}$$

Next, we have three spaces of dimension $n - 1$ which are isomorphic to $\mathbf{V}(n)$ from equation 2. The first space is the space $\mathcal{V}_2$ of diagonal matrices whose diagonal sums to zero and the next two $n - 1$ dimensional spaces are the space of matrices whose rows (respectively columns) are constant, and columns (respectively rows) sum to zero:

$$\mathcal{V}_3 = \{r1_n^T | \sum_{i=1}^{n} r_i = 0\}, \quad \mathcal{V}_4 = \{1_n c^T | \sum_{i=1}^{n} c_i = 0\}$$

Finally, there are two larger irreducible spaces:

$$\mathcal{V}_5 = \{A| \quad A = -A^T, A1_n = 0_n\}, \quad \mathcal{V}_6 = \{A| \quad A = A^T, A1_n = 0_n, A_{ii} = 0, \forall i = 1, \ldots, n\}$$

The dimension of $\mathcal{V}_5$ is $\binom{n-1}{2} = \frac{n^2-3n}{2} + 1$, and the dimension of $\mathcal{V}_6$ is $\binom{n-1}{2} - 1 = \frac{n^2-3n}{2}$. So, we have the following decomposition:

$$\mathcal{V} = \mathcal{V}_0 \oplus \mathcal{V}_1 \oplus \mathcal{V}_2 \oplus \mathcal{V}_3 \oplus \mathcal{V}_4 \oplus \mathcal{V}_5 \oplus \mathcal{V}_6 \cong \mathcal{V}_0^{\oplus 2} \oplus \mathcal{V}_3^{\oplus 3} \oplus \mathcal{V}_5 \oplus \mathcal{V}_6$$

Accordingly, using Schur's lemma, each linear mapping can be characterized by $3^2 + 2^2 + 1 + 1 = 15$ parameters, the same result from Maron et al. (2018), so we have the expected number of parameters. In appendix C, we give a formal proof of these arguments and present an algorithm to decompose an input matrix $\mathbf{X}$ into its seven irreducible spaces with linear complexity in the matrix size $n^2$. These results are summarized in the following theorem.

**Theorem 4.1.** *For all $n \geq 4$, the space $\mathbb{R}^{n \times n}$ can be written as a direct sum of the spaces $\mathcal{V}_0, \ldots, \mathcal{V}_6$. These spaces are invariant and irreducible, and the isomorphism relations between them are given by $\mathcal{V}_0 \cong \mathcal{V}_1, \mathcal{V}_2 \cong \mathcal{V}_3 \cong \mathcal{V}_4$.*

### 4.3 DEEP WEIGHT SPACES

Recently, there has been growing interest in devising neural operators: neural networks that operate on an input, which is itself a neural network. This type of problem is of interest for various tasks involving post-processing and synthesis of multiple trained neural networks, as well as for processing Implicit Neural Representations (INRs), which are a popular alternative for representing certain standard data structures. (see e.g. Kalogeropoulos et al. (2024) for further discussion).

A key concept in the design of 'neural operators' has been the requirement that they are equivariant to the input neural data (Kalogeropoulos et al., 2024; Kofinas et al., 2024; Zhou et al., 2024a; 2023; Lim et al., 2024) In this section we consider the setting discussed in Navon et al. (2023); Zhou et al. (2024b), where the neural data is a collection of weights and biases $(W_m, b_m)_{m=1}^M$ representing an MLP of depth $M$, and a neural operator is constructed by composing standard activation with linear mappings which are equivariant with respect to the multi-permutation action we will now describe:

The output of an MLP architecture is invariant to the permutation of its hidden neurons. For example, the MLP defined by $W_2 \cdot ReLU \cdot (W_1 x)$ will remain the same function if we replace weights $W_2, W_1$ with the weights $W_2 P, P^T W_1$.

To define the symmetries of learning on MLPs in full generality, we will adapt the notation from Navon et al. (2023). We consider the space of MLP parameters with a given fixed depth $M$ and layer dimensions $d_0, \ldots, d_{M+1}$ ($M$ and all $d_j$ are assumed to be larger than one). These are parameterized by the vector space $\mathcal{V} = \bigoplus_{m=1}^M (\mathcal{W}_m \oplus \mathcal{B}_m)$ where $\mathcal{W}_m := \mathbb{R}^{d_m \times d_{m-1}}$ and $\mathcal{B}_m := \mathbb{R}^{d_m}$ represent the weights and biases of the $m$-th layer. The symmetry group of the weight space is the direct product of symmetric groups $\mathcal{G} = S_{d_1} \times \cdots \times S_{d_{M-1}}$. An element $g = (\tau_1, \ldots, \tau_{M-1})$ in the group acts on an element $v = [W_m, b_m]_{m \in [M]}$ from $\mathcal{V}$ as follows:

$$\rho(g)v = [W'_m, b'_m]_{m \in [M]}, \tag{4a}$$

$$W'_1 = P_{\tau_1}^T W_1, \ b'_1 = P_{\tau_1}^T b_1, \tag{4b}$$

$$W'_m = P_{\tau_m}^T W_m P_{\tau_{m-1}}, \ b'_m = P_{\tau_m}^T b_m, \ m \in [2, M-1] \tag{4c}$$

$$W'_M = W_M P_{\tau_{M-1}}, \ b_{M'} = b_M. \tag{4d}$$

where $P_{\tau_m} \in \mathbb{R}^{d_m \times d_m}$ is the permutation matrix of $\tau_m \in S_{d_m}$.

Previous work (Navon et al., 2023; Zhou et al., 2024b) has already characterized all linear equivariant functions from $\mathcal{V}$ to itself. However, this characterization requires tedious bookkeeping and division into a large number of different cases. Here we show how to decompose $\mathcal{V}$ into irreducibles in a rather straightforward way, and as a result obtain a (arguably) simpler characterization of all linear equivariant layers.

We claim that $\mathcal{V}$ is a direct sum of multiple copies of $2M - 3$ irreducible representations of $\mathcal{G}$:

 1. The first irreducible representation is the trivial scalar representation $\mathbf{S} = \mathbb{R}$ with the trivial action $gx = x$.

 2. The next $M - 1$ representations are the vector representations $\mathbf{V}_m, m = 1, \ldots, M - 1$ of vectors in $\mathbb{R}^{d_m}$ which sum to zero (the spaces $\mathbf{V}(d_m)$ from equation 2), with the action of $g$ being permutation of the vector by the $m$-th permutation $\tau_m$.

3. The last $M - 2$ representations are representations $\mathbf{M}_m, \quad m = 2, \ldots, M - 1$ of $d_m \times d_{m-1}$ matrices whose rows and columns sum to zero, where the action of $g$ on a matrix $W$ in this space is given by $P_{\tau_m}^T W P_{\tau_{m-1}}$.

The linear equivariant layers from $\mathcal{V}$ to itself can be inferred from the following theorem:

**Theorem 4.2.** *The spaces* $\mathbf{S}, \mathbf{V}_1, \ldots, \mathbf{V}_{M-1}, \mathbf{M}_2, \ldots, \mathbf{M}_{M-1}$ *are absolutely irreducible, and are not isomorphic to each other.* $\mathcal{V}$ *is isomorphic to* $\mathbf{S}^{\oplus \alpha} \oplus \left( \oplus_{m=1}^{M-1} \mathbf{V}_m^{\oplus \beta_m} \right) \oplus \left( \oplus_{m=2}^{M-1} \mathbf{M}_m \right)$, *where*

$$\alpha = d_0 + 2d_M + 2M - 3, \quad \beta_1 = d_0 + 2, \quad \beta_{M-1} = d_M + 2, \quad \beta_m = 3, \forall m = 2, \ldots, M - 2 \quad (5)$$

*In particular, the space of equivariant linear mappings from* $\mathcal{V}$ *to itself is thus of dimension* $\alpha^2 + \sum_{m=1}^{M-1} \beta_m^2 + (M - 2)$.

*Proof.* The spaces $\mathbf{S}, \mathbf{V}_1, \ldots, \mathbf{V}_{M-1}, \mathbf{M}_2, \ldots, \mathbf{M}_{M-1}$ are not isomorphic (even when they have the same dimension). The action of $\mathcal{G}$ on $\mathbf{S}$ is trivial while the action on the other spaces is not. The other spaces aren't isomorphic to each other since different components of the multi-permutation $g \in \mathcal{G}$ acts on each space. The spaces $\mathbf{S}$ and $\mathbf{V}_m$ are absolutely irreducible, as we discussed in Subsection 4.1. We will explain why $\mathbf{M}_m$ are absolutely irreducible in Appendix D.

Obtaining the decomposition of $\mathcal{V}$ is rather straightforward. First, following Navon et al. (2023), we identify the weight spaces $\mathcal{W}_m$ and bias spaces $\mathcal{B}_m$ with subspaces $\hat{\mathcal{W}}_m$ and $\hat{\mathcal{B}}_m$ of $\mathcal{V}$ by zero padding, and note that each one of these subspaces is $\mathcal{G}$-invariant and that their direct sum gives us the full parameter space $\mathcal{V}$. We can then decompose each one of these spaces into irreducibles to obtain

$$\hat{\mathcal{B}}_m \cong \mathbf{S} \oplus \mathbf{V}_m, \quad m = 1, \ldots, M - 1 \tag{6a}$$

$$\hat{\mathcal{B}}_M \cong \mathbf{S}^{\oplus d_M} \tag{6b}$$

$$\hat{\mathcal{W}}_1 \cong \mathbf{S}^{\oplus d_0} \oplus \mathbf{V}_1^{\oplus d_0} \tag{6c}$$

$$\hat{\mathcal{W}}_m \cong \mathbf{S} \oplus \mathbf{V}_{m-1} \oplus \mathbf{V}_m \oplus \mathbf{M}_m, \quad m = 2, \ldots, M - 1 \tag{6d}$$

$$\hat{\mathcal{W}}_M \cong \mathbf{S}^{\oplus d_M} \oplus \mathbf{V}_{M-1}^{\oplus d_M} \tag{6e}$$

The multiplicities of each irreducible in $\mathcal{V}$, specified in equation 5, can now be found by simply counting how many times each irreducible appears in the decomposition above.

The decomposition in equation 6 can actually be easily obtained from the 'deepsets' decomposition of the $S_n$ on $\mathbb{R}^n$ described previously: the action of $\mathcal{G}$ on $\hat{\mathcal{B}}_m$ with $m < M$ is isomorphic to the action of $S_{d_m}$ on $\mathbb{R}^m$ and hence we get the exact 'DeepSets' decomposition from Section 4.1. The action of $\mathcal{G}$ on $\hat{\mathcal{B}}_M$ is trivial and hence can be written as a direct sum of $d_M$ trivial one dimensional $\mathbf{S}$ spaces. The action on $\hat{\mathcal{W}}_1$ (and $\hat{\mathcal{W}}_M$) multiplies a matrix by a permutation from the left (right), and hence can be seen as a direct sum of 'deep-sets' actions on the columns (rows) of the matrix. Finally, the action of $\mathcal{G}$ on $\hat{\mathcal{W}}_m$ for $m = 2, \ldots, M - 1$ is a tensor product of the natural action of $S_{d_{m-1}}$ on $\mathbb{R}^{d_{m-1}}$ and the action of $S_{d_m}$ on $\mathbb{R}^{d_m}$, and therefore its irreducible decomposition can be obtained by taking tensor products of the irreducible representations of $\mathbb{R}^{d_m}$ and $\mathbb{R}^{d_{m-1}}$, as explained in more detail in the full proof. $\qquad \square$

We note that equation 6 includes almost all information required to compute linear equivariant maps from $\mathcal{V}$ to itself. All is left is the decomposition algorithm to write each $(W_m, b_m)_{m \in [M]}$ as a direct sum of elements in the irreducible decomposition. This can be done independently for each subspace $\mathcal{W}_m$ and $\mathcal{B}_m$. The decomposition for $\hat{\mathcal{W}}_m$ in equation 6d is not immediate and we will explain it in Appendix D.1.

The decomposition in equation 6 and equation 5 also provides substantial additional information. For example, we can immediately see that there are $\alpha$ invariant maps from $\mathcal{V}$ to $\mathbb{R} = \mathbf{S}$, which correspond to the number of copies of the trivial representation $\mathbf{S}$ in $\mathcal{V}$. Moreover, if we are interested in the equivariant maps from a bias space $\hat{\mathcal{B}}_i$ (or weight space $\hat{\mathcal{W}}_i$) to another bias space $\hat{\mathcal{B}}_j$ (or weight space $\hat{\mathcal{W}}_j$), we can easily infer the equivariant mappings from the decomposition in equation 6. For example, when $i = j = M$ there will be $d_M^2$ mappings since $\hat{\mathcal{B}}_m$ consists of $d_M$ isomorphic

representations, and when $i < j = M$ there will be $d_M$ mappings since $\mathbf{S}$ appears a single time in $\hat{\mathcal{B}}_i$ and $d_M$ times in $\hat{\mathcal{B}}_M$. Continuing in this way we can reconstruct all the different bias-to-bias, bias-to-weight, weight-to-bias, and weight-to-weight cases analyzed in Tables 5-8 of Navon et al. (2023), and Tables 8-11 in Zhou et al. (2024b). More importantly, these tables which were necessary for implementing weight space layers in previous work, are not necessary when implementing these layers using Schur's lemma as we suggest.

## 5  SETS OF UNALIGNED SYMMETRIC ELEMENTS

Next, we consider the setting where our data is a $k$-tuple of 'unaligned objects' $(v_1, \dots, v_k)$, each coming from a representation $\mathcal{V}$ of $\mathcal{G}$, and we want to learn functions which are equivariant with respect to the joint action of a $k$-tuple of group elements $(g_1, \dots, g_k)$ on each coordinate independently, and to a permutation $\tau \in S_k$ of the $k$-tuple.

We define $\mathcal{G} \wr S_k := \mathcal{G} \times \cdots_{k-times} \times \mathcal{G} \times S_k$, and the action is given by

$$(\tau, g_1, .., g_k) \cdot (v_1, .., v_k) = (g_{\tau(1)} \cdot v_{\tau(1)}, ..., g_{\tau(k)} \cdot v_{\tau(k)}) \tag{7}$$

The group for which this action is defined is also called the restricted wreath product of $\mathcal{G}$ and $S_k$. For more details, see Wang et al. (2020).

This type of 'wreath-equivariant-structure' arises in several settings. One is 'alignment problems', where our goal is, given a pair of elements $(v_1, v_2) \in \mathcal{V}^k, k = 2$, to find the group element $g^* = g^*(v_1, v_2)$ which makes $v_1$ 'as similar as possible' to $v_2$. This task is equivariant to application of $\mathcal{G}$ elements to each coordinate because if $g^* v_1 \approx v_2$ then $g_2 g^* g_1^{-1}(g_1 v_1) \approx g_2 v_2$. Similarly, this task is equivariant to permuting $v_1$ and $v_2$, because if $g^* v_1 \approx v_2$ then $(g^*)^{-1} v_2 \approx v_1$. For a more detailed derivation, see Chen & Wang (2024); Navon et al. (2024), which discussed these problems for sets and weight spaces, respectively. Additional examples of 'wreath-equivariant-problems' are the anomaly detection problem discussed in the experimental section, and problems with hierarchical structures as discussed in Wang et al. (2020).

**Wreath-equivariant layers.** Our aim is to characterize all $\mathcal{G} \wr S_k$ equivariant mappings from $\mathcal{V}^k$ to itself. We note that any linear $\mathcal{G}$-equivariant mapping $\hat{L} : \mathcal{V} \to \mathcal{V}$ induces a 'Siamese' $\mathcal{G} \wr S_k$ equivariant mapping defined by

$$L(v_1, \dots, v_k) = (\hat{L}(v_1), \dots, \hat{L}(v_k)).$$

The interesting question is how many additional mappings are present. This problem was previously studied in Maron et al. (2020); Wang et al. (2020) when $\mathcal{G}$ is a finite group acting on $\mathbb{R}^n$ *transitively* by permutations (this means that the action of $\mathcal{G}$ on $[n]$ has a single orbit). In this setting, the equivariant mappings are composed of the Siamese mappings and a single additional non-Siamese mapping. However, the transitivity assumption does not hold in many examples of interest, such as the graph and weight space examples discussed in this paper. In our analysis, we will release the transitivity assumption, and allow $\mathcal{G}$ to be a general finite group. In some cases, this will lead to a substantial number of non-Siamese networks.

To characterize $\mathcal{G} \wr S_k$ equivariant functions, we first aim to characterize all invariant irreducible sub-spaces of $\mathcal{V}^k$, assuming we know all irreducible sub-spaces of $\mathcal{V}$. An important role will be played by *trivial representations*: representations $\mathbf{S}$ of $\mathcal{G}$ such that $gv = v$ for all $g \in \mathcal{G}$ and $v \in \mathbf{S}$.

**Theorem 5.1.** *Let $\mathcal{V}$ be a real representation of $\mathcal{G}$, with irreducible decomposition*

$$\mathcal{V} = (\oplus_{i=1}^s \mathbf{S}_i) \oplus (\oplus_{j=1}^t \mathcal{V}_t) \tag{8}$$

*where $\mathbf{S}_i$ are trivial representations and $\mathcal{V}_t$ are not. Then an irreducible decomposition for $\mathcal{V}^k$ with respect to the action of $\mathcal{G} \wr S_k$ is given by*

$$\mathcal{V}^k = (\oplus_{i=1}^s \mathbf{S}_{i,0}^k) \oplus (\oplus_{i=1}^s \mathbf{S}_{i,1}^k) \oplus (\oplus_{j=1}^t \mathcal{V}_t^k),$$

$$where \quad \mathbf{S}_{i,0}^k = \{(s_1, \dots, s_k) \in \mathbf{S}_i^k, \sum_{i=1}^k s_i = 0\}, \quad \mathbf{S}_{i,1}^k = \{(s, \dots, s) \in \mathbf{S}_i^k\}$$

*Proof.* The fact that $\mathcal{V}_i$ is $\mathcal{G}$ invariant implies easily that $\mathcal{V}_i^k$ is $\mathcal{G} \wr S_k$ also invariant. In the appendix E.1, we will show that if the action of $\mathcal{G}$ on $\mathcal{V}$ is not trivial then $\mathcal{V}^k$ is irreducible representation of $\mathcal{G} \wr S_k$. In contrast, for the spaces $\mathbf{S}_i^k$ the action of $\mathcal{G}$ is trivial, so this representation can be identified with the standard representation $\mathbb{R}^k$ of $S_k$. The decomposition to $\mathbf{S}_{i,0}^k$ and $\mathbf{S}_{i,1}^k$ then follows from the 'DeepSets decomposition' discussed in Section 4.1. □

To count the number of linear equivariant mappings $L : \mathcal{V}^k \to \mathcal{V}^k$, we note that

$$\mathbf{S}_{i,0}^k \cong \mathbf{S}_{j,0}^k, \quad \mathbf{S}_{i,1}^k \cong \mathbf{S}_{j,1}^k, \quad \mathbf{S}_{i,0}^k \ncong \mathbf{S}_{i,1}^k, \quad \forall 1 \le i < j \le k$$

Thus, the number of linear equivariant mappings from the 'trivial part' of $\mathcal{V}^k$, that is $\left( \oplus_{i=1}^s \mathbf{S}_{i,0}^k \right) \oplus \left( \oplus_{i=1}^s \mathbf{S}_{i,1}^k \right)$, to itself, is $2s^2$, while the number of linear equivariant mappings from the 'trivial part' of $\mathcal{V}$ to itself (which is equal to the number of Siamese layers), is $s^2$. As a result, we obtain $s^2$ non-Siamese $\mathcal{G} \wr S_k$ equivariant maps.

**Examples.** We found that the number of non-Siamese layers is $s^2$, where $s$ is the number of trivial representations in $\mathcal{V}$. Let us consider the implications for the three examples we have discussed earlier:

1. **Deep-sets.** In the DeepSets setting where $\mathcal{V} = \mathbb{R}^n, \mathcal{G} = S_n$, there is a single trivial representation of constant vectors, and therefore in this setting there is a unique non-Siamese layer for $\mathcal{V}^k$. Indeed, this is the case for any group acting transitively by permutations on $\mathbb{R}^n$, and thus we obtain the results for transitive actions from Maron et al. (2020); Wang et al. (2020).

2. **Graphs.** In the graph setting where $\mathcal{V} = \mathbb{R}^{n \times n}, \mathcal{G} = S_n$ there are $s = 2$ trivial representations (see equation 3), and therefore $s^2 = 4$ non-Siamese layers.

3. **Weight Spaces.** In the weight space examples the number $s$ of trivial representations is rather large, $\alpha = d_0 + 2d_M + 2M - 3$ from equation 5, and there are $\alpha^2$ non-Siamese mappings.

In the next theorem we give an explicit characterization of the non-Siamese layers of $\mathcal{V}^k$. For this characterization, we note that the direct sum of all trivial representations of $\mathcal{V}$ (as in equation 8) is the vector space $\mathcal{V}_{fixed} = \{v \in \mathcal{V} | gv = v, \forall g \in \mathcal{G}\}$. Alternatively, every basis $e_1, \ldots, e_s$ of $\mathcal{V}_{fixed}$ defines a decomposition of $\mathcal{V}_{fixed}$ into one-dimensional irreducible invariant sub-spaces $\mathcal{S}_i = \{ce_i | c \in \mathbb{R}\}$. Accordingly, our characterization of non-Siamese layers is based on such bases:

**Theorem 5.2.** *Let $\mathcal{V}$ be a real representation of a finite group $\mathcal{G}$. and let $e_1, \ldots, e_s$ be a basis to the subspace $\mathcal{V}_{fixed}$. Let $\langle \cdot, \cdot \rangle$ be a $\mathcal{G}$ invariant inner product on $\mathcal{V}$. Then every linear equivariant map $L : \mathcal{V}^k \to \mathcal{V}^k$ is of the form*

$$L(v_1, \ldots, v_k) = \sum_{i,j=1}^s a_{ij} \left( \sum_{\ell=1}^k \langle v_\ell, e_i \rangle e_j, \ldots, \sum_{\ell=1}^k \langle v_\ell, e_i \rangle e_j \right) + \left( \hat{L}(v_1), \ldots, \hat{L}(v_k) \right) \quad (9)$$

*where $\hat{L} : \mathcal{V} \to \mathcal{V}$ is a linear equivariant map, and $a_{ij}$ are real numbers. Conversely, every linear mapping of the form defined in equation 9 is equivariant.*

The theorem is proven in Appendix E. We note that an invariant inner product on $\mathcal{V}$ is an inner product satisfying $\langle gv, gu \rangle = \langle v, u \rangle$ for all $g \in \mathcal{G}$ and $u, v \in \mathcal{V}$. When $\mathcal{G}$ is finite, an invariant inner product always exists: it can be obtained by starting from an arbitrary inner product and then averaging over the group (Fulton & Harris, 2013). In the examples we consider in this paper, the group $\mathcal{G}$ acts by permutations. In this case, the standard $\ell_2$ inner product is invariant. Implementing the non-Siamese layers defined in equation 5.2 only requires finding a basis for $\mathcal{V}_{fixed}$. In particular, if $\mathcal{V}$ is one of the spaces discussed previously, the basis $e_i$ is just a choice of an element from each of the trivial spaces $\mathbf{S}_i$ in the decomposition of $\mathcal{V}$.

**Beyond $S_k$.** So far, we have considered the action of $\mathcal{G} \wr S_k$ where $\mathcal{G}$ is finite. We can also generalize these results to the setting $\mathcal{G} \wr H$, where $H$ is a subgroup of $S_k$ which acts transitively on $\{1, \ldots, k\}$. By doing this, we are generalizing the results in Wang et al. (2020), which assumes that both $\mathcal{G}$ and $\mathcal{H}$ act transitively.

The full generalization is described in Appendix E.1. The general idea is this: Since $\mathcal{G}$ acts trivially on $\mathcal{V}_{fixed}$, the action of $\mathcal{G} \wr H$ on $\mathcal{V}_{fixed}^k$ can be identified with the action of $H$ on $\mathcal{V}_{fixed}^k$. In turn, $\mathcal{V}_{fixed}^k$ is a direct sum of $s$ different copies of $\mathbb{R}^k$. Accordingly, the number of $H$ equivariant maps from $\mathcal{V}_{fixed}^k$ to itself is $s^2 \cdot h$, where $h$ is the number of $H$ equivariant maps from $\mathbb{R}^k$ to itself. One of these $h$ maps is the identity map, which is Siamese; therefore, the total number of non-Siamese equivariant maps is $(h-1) \cdot s^2$. In the case where $H = S_k$, we have that $h = 2$, and therefore, in our analysis above, we have found $s^2$ non-Siamese maps.

## 6 EXPERIMENTS

In this section, we consider problems with a wreath-equivariant structure. We compare networks implementing the complete basis of equivariant layers that we have found, with networks that only use Siamese layers or combine a partial list of non-Siamese layers suggested in previous works. Implementation details of all experiments are described in Appendix A.

**Graph Anomaly Detection.** We begin with a synthetic wreath-equivariant task in which Siamese networks will fail by design: we consider a graph anomaly detection problem, where the input is $k$ graphs with $n$ nodes, $(\mathbb{G}_1, \ldots, \mathbb{G}_k)$, where most graphs are similar, and one is an 'anomaly.' The output is a $k$ dimensional probability vector, where the $i$-th entry denotes the probability that $\mathbb{G}_i$ is an anomaly. This task is $\mathcal{G} \wr S_k$ equivariant (where $\mathcal{G} = S_n$), where the action on the input space is as in equation 7, and the action on the output space is just permuting the entries of the probability vector. We generate data for this problem as follows: We randomly generate two graphs $\mathbb{G}$ and $\hat{\mathbb{G}}$ using the Erdos-Reyni distribution. We take $k-1$ copies of the graphs $(\mathbb{G}_1, \ldots, \mathbb{G}_{k-1})$ to be permuted copies of $\mathbb{G}$, and one of them to be $\hat{\mathbb{G}}$ and insert it in a random location. We also add some noise with variance $\eta$ to all graphs.

We consider equivariant models for this task, composed of several linear wreath-equivariant layers and point-wise ReLU activations. The final layer is a point-wise summation of each of the $k$ graphs to obtain a final vector in $\mathbb{R}^k$, followed by a softmax layer to obtain a probability vector. For the linear wreath-equivariant layers, we consider several alternatives: Siamese layers only, adding the single additional non-Siamese layer suggested in Wang et al. (2020); Maron et al. (2020) denoted by DSS, and the full model we suggest, which has four non-Siamese layers (we name our model SchurNet). The Siamese layers are implemented using the decomposition computed in Subsection 4.2.

The results of this experiment are shown in Table 1. As we can see, Siamese networks attain 10% accuracy without noise (and even lower accuracy with noise). This is to be expected since Siamese features can be useful to differentiate between $\mathbb{G}$ and $\hat{\mathbb{G}}$, but not to determine how many times each one of them occurs in a $k$-tuple.

| Model/Noise | $\eta = 0.0$ | $\eta = 0.1$ |
|---|---|---|
| Siamese | 10% | 10% |
| DSS | 97.5% | 92% |
| SchurNet (Ours) | **100%** | **97.0%** |

Table 1: Performance comparison of models at different noise levels ($\eta$).

In contrast, networks with non-Siamese layers attain significantly better performance, whereas our method, which contains the maximal number of non-Siamese layers, attains the best performance.

**Wasserstein Distance Computation** We next consider the task of learning the Wasserstein distance, as discussed in Amir & Dym (2024); Chen & Wang (2024); Haviv et al. (2024). The Wasserstein distance between two unordered multisets of vectors in $\mathbb{R}^d$, denoted by $\{x_1, \ldots, x_n\}$ and $\{y_1, \ldots, y_n\}$, is defined to be the minimal distance between the sets, under the optimal permutation giving the best alignment between them (as in the alignment problem discussed in the beginning of Section 5). Comput-

| Dataset | Input | SchurNet | NProductNet |
|---|---|---|---|
| noisy-sphere-3 | [100, 300] | **0.0389** | 0.046 |
| | [300, 500] | **0.1026** | 0.158 |
| noisy-sphere-6 | [100, 300] | 0.0217 | **0.015** |
| | [300, 500] | 0.0795 | **0.049** |
| uniform | 256 | 0.0974 | **0.097** |
| | [200, 300] | **0.1043** | 0.1089 |
| ModelNet-small | [20, 200] | **0.0623** | 0.084 |
| | [300, 500] | **0.0738** | 0.111 |
| ModelNet-large | 2048 | **0.0468** | 0.140 |
| | [1800, 2000] | **0.0551** | 0.162 |
| RNAseq | [20, 200] | 0.0123 | **0.012** |
| | [300, 500] | **0.0334** | 0.292 |

Table 2: Comparison of SchurNet and NProductNet.

ing Wasserstein distances can be computationally intensive, so this task aims to devise a neural network to learn the distance instead. Learning Wasserstein distances is a $S_n \wr S_2$ invariant task. In Chen & Wang (2024), a Siamese approach named NProductNet is suggested to address this problem. We compare SchurNet to Chen & Wang (2024) to demonstrate that adding our non-siamese layers enhances the performance of standard Siamese layers. We note that Amir & Dym (2024) achieved state-of-the-art results for this task with a very different method, which does not follow the conventional paradigm of a stack of linear layers and non-linear activation functions. Therefore, we have no direct way of adding our non-siamese layers to their method for comparison. In our experiments, we add the additional $d^2$ non-Siamese layers to their implementation and compare performance on the datasets addressed in their paper. The results, reported in Table 2, show that adding non-Siamese layers improves performance in most cases. We note that each dataset has two options: the top option in each table row depicts the case where training and test distributions had the same dimension, while the bottom option checks the generalization to test distribution of different sizes.

**Weight Space Alignment.** We consider the alignment task discussed above in the setting where $v_1, v_2$ are elements in weight spaces, and the task is to find the group element that optimally aligns $v_1$ and $v_2$. One interesting application of this problem is model merging: in Ainsworth et al., it was shown that linear interpolation of $v_1, v_2$ gives a new network whose performance is considerably worse, but interpolation after alignment leads to much better results.

The weight space alignment problem was considered in Ainsworth et al.; Tatro et al. (2020); Peña et al. (2023). Recently, Navon et al. (2024) outperformed these methods using a learning approach based on the DWS layers from Navon et al. (2023), applied to $v_1, v_2$ in a Siamese fashion. This experiment aims to check whether adding non-Siamese layers improves their results. We take only part of the described layers in Theorem 5.2 layers: the mappings from each weight/bias space to itself, excluding, for example, non-Siamese weight to other weight or weight to bias. This added fewer parameters and generalized better than taking all mappings.

| Model | MNIST | | CIFAR10 | |
|---|---|---|---|---|
| | Acc($\downarrow$) | Loss ($\downarrow$) | Acc($\downarrow$) | Loss ($\downarrow$) |
| **SchurNet (Ours)** | 1.5e-5 | **0.251346** | **0.0** | **1.7822** |
| **Siamese** | **1.25e-5** | 0.262913 | 1.0e-4 | 1.7876 |

Table 3: Comparison of SchurNet and Siamese models on MNIST and CIFAR10.

We run two sets of experiments from Navon et al. (2024): one on MLPs trained on MNIST (LeCun et al., 1998) and one on MLPs trained on the CIFAR10 (Krizhevsky et al., 2009) dataset. We trained both models for 100 epochs and reported the test accuracy and the reconstruction loss. We ran several hyper-parameter configurations for both methods to ensure a fair comparison and reported the results with the best reconstruction loss. As we can see, our model with non-Siamese layers outperforms the Siamese network from Navon et al. (2024) in all settings.

**Conclusion.** In this paper, we revisited the idea of using irreducible representations instead of parameter-sharing to characterize equivariant linear layers for representations of permutations and related groups. Using this approach, we obtained alternative derivations for the characterizations of equivariant layers from DeepSets and 2-IGN, and a significantly simplified derivation of deep weight spaces equivariant layers. We have also obtained a previously unknown characterization for the linear equivariant layers of wreath products $\mathcal{G} \wr S_n$, and showed the benefits of using the full characterization for several wreath-equivariant tasks. In general, looking forward to other yet unknown applications, what could be the benefit of using the irreducible approach? One answer is reduced book-keeping, as exemplified in the deep weight space example. In general, the number of equivariant layers can be, in the worst case, quadratic in the number of irreducibles. When this happens, the irreducible approach is expected to lead to simpler characterizations that are easier to implement. Additionally, networks like $k$-IGN (Maron et al., 2018), which are based on tensor representations, use intermediate features in $n^k$, and using irreducibles could lead to new equivariant models with intermediate irreducible features of lower dimensions. Finding the irreducible decompositions of $k$-IGN when $k > 2$ and the potential applications of this decomposition is an interesting avenue for future work.

## 7 ACKNOWLEDGMENTS

We would like to extend our thanks to Eitan Rosen for the valuable discussions. We also sincerely thank Idan Tankel for his technical support, as without your help, none of this would have worked as it has. The authors are supported by ISF grant 272/23.

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

## A    IMPLEMENTATION DETAILS

In this section, we detail the hyper-parameters used through learning. In all our models we used Adam or AdamW optimizers.

**Graph Anomaly Detection**    In this set of experiments we run each model with 27 different hyper-parameters and reported the best for each model. We run through different learning rates, weight decays and learning rate decay. In our model, we used four layers, each with a hidden dimension of 256 and ReLU activation.

**Deep weight space alignment**    In this section we took the model of Navon et al. (2024) and added our non-Siamese common layers. We searched for each model and dataset over four hyper-parameters and for each reported the best performance. We trained for 100 epochs and reported the reconstruction loss.

**Wasserstein Distance computation**    In this set we took the model of Chen & Wang (2024) and added the non-Siamese layers. We trained for 200 epochs and reported the relative absolute mean error. In most of the experiments we used learning rate of $1e^{-4}$ and weight decay of $1e^{-5}$.

## B    ABSOLUTELY IRREDUCIBLE REAL REPRESENTATIONS

Here we give some more details on the concept of absolutely irreducible real representations.

As discussed in the main text, a real irreducible representation $\mathcal{V}$ is called *absolutely irreducible* if its complexification is irreducible over $\mathbb{C}$. The complexification is denoted by $c\mathcal{V}$. It can be defined by choosing a basis $v_1, \ldots, v_n$ to $\mathcal{V}$ and then defining

$$c\mathcal{V} = \{\sum_{i=1}^{n} c_i v_i, \quad c_i \in \mathbb{C}\}.$$

Let $T : \mathcal{V} \to \mathcal{V}$ be an equivariant non-zero mapping, and assume $\mathcal{V}$ is absolutely irreducible. Then $Tv = \lambda v$ for some real $\lambda$. This is because $T$ can be linearly extended to a non-zero linear mapping $T : c\mathcal{V} \to c\mathcal{V}$. Similarly, the group action of $\mathcal{G}$ which is linear can also be extended to $c\mathcal{V}$. The extension of $T$ to $c\mathcal{V}$ will be equivariant, and therefore, since $c\mathcal{V}$ is irreducible by assumption, Schur's Lemma for complex representations implies that $T = \lambda I$ for $\lambda \in \mathbb{C}$. Since $T\mathcal{V} = \mathcal{V}$, we know $\lambda$ must be real.

**Real irreducible permutation representations are absolutely irreducible.**    Classical representation theory results (see e.g., Fulton & Harris (2013)) characterize all complex irreducible representation of the permutation group, up to isomorphism. In this process, they show that all these representations can be defined over the rational numbers. This means that, for any complex irreducible representation $\mathcal{U}$ of the permutation group $S_n$, there exists a basis in which for every $g \in \mathcal{G}_n$, the matrix representing the linear mapping defined by $g$ will be rational.

As explained, e.g., in Webster, this fact implies that every irreducible $s$ dimensional real representation $\mathcal{V}$ is absolutely irreducible. Indeed, if $c\mathcal{V}$ were not irreducible over $\mathbb{C}$, then $c\mathcal{V} = \mathcal{V}_1 \oplus \mathcal{V}_2$ where $\mathcal{V}_1$ is an invariant (complex) irreducible subspace, and the dimension $t$ of $\mathcal{V}_1$ satisfies $0 < t < s$. In an appropriate bases $u_1, \ldots, u_t$ of $\mathcal{V}_1$ we will have that the matrix representing each $g \in S_n$ is rational. We can write each $u_j$ as

$$u_j = \sum_{k=1}^{n}(a_{jk} + ib_{jk})v_k.$$

One can then verify that the $t$ dimensional real space $\hat{\mathcal{V}}_1$ defined by

$$\hat{u}_j = \sum_{k=1}^{n} a_{jk}v_k$$

is a (real) $\mathcal{G}$ invariant subspace of $\mathcal{V}$ and has dimension $t < s$, which contradicts the fact that $\mathcal{V}$ is irreducible.

## C GRAPH NETWORKS IRREDUCIBLE DECOMPOSITION

In the main text, we claim that, for $n \geq 4$, the representation $\mathbb{R}^{n \times n}$ of the permutation group $S_n$ can be decomposed into a direct sum of seven irreducible representations:

$$\mathcal{V}_0 = \{a \cdot I_n | a \in \mathcal{R}\}$$
$$\mathcal{V}_1 = \{a \cdot (1_{n \times n} - I_n) | a \in \mathcal{R}\}$$
$$\mathcal{V}_2 = \{Diag(\alpha_1, .., \alpha_n) | \sum_{i=1}^{n} \alpha_i = 0\}$$
$$\mathcal{V}_3 = \{r 1_n^T | \sum_{i=1}^{n} r_i = 0\}$$
$$\mathcal{V}_4 = \{1_n c^T | \sum_{i=1}^{n} c_i = 0\}$$
$$\mathcal{V}_5 = \{A | A = -A^T, A 1_n = 0\}$$
$$\mathcal{V}_6 = \{A | A = A^T, A 1_n = 0, A_{ii} = 0, \forall i = 1, \ldots, n\}.$$

We begin by giving an algorithm describing how every $n$ by $n$ matrix can be decomposed into a sum of matrices from these spaces, with a computational complexity of $O(n^2)$. This is necessary for a 2-IGN implementation which is based on irreducibles, and is also a first step towards a full proof of Theorem 4.1. This full proof will be presented after the algorithm.

### C.1 DECOMPOSITION ALGORITHM

We will give an algorithm to decompose a matrix $A \in \mathbb{R}^{n \times n}$ into a sum of matrices from these seven sub-spaces. We assume $n \geq 4$ (actually, when $n = 3$ the algorithm works as well, but in this case $\mathcal{V}_6$ is zero dimensional as discussed in the main text).

The first step of the algorithm is to write

$$A = B + \begin{pmatrix} 0 & a & \ldots & a \\ a & 0 & \ldots & a \\ \vdots & & & \\ a & a & \ldots & 0 \end{pmatrix} + \begin{pmatrix} b & & & \\ & b & & \\ \vdots & & & \\ & & \ldots & b \end{pmatrix}$$

where the two matrices from the right are in $\mathcal{V}_0$ and $\mathcal{V}_1$, and $a$ and $b$ are the average of off-diagonal and diagonal elements of $A$, respectively, so that the diagonal and off-diagonal elements of $B$ both sum to zero. It remains to find a decomposition for $B$. The crucial part of this is the following lemma

**Lemma C.1.** *Let $n > 2$ be a natural number, and let $B \in \mathbb{R}^{n \times n}$ be a matrix whose diagonal elements sum to zero, and off-diagonal elements sum to zero. Then there exists a matrix $C$ in*

$$\hat{\mathcal{V}} = \{C \in \mathbb{R}^{n \times n} | \quad , C 1_n = 0, C^T 1_n = 0 \text{ and } C_{ii} = 0, \forall i = 1, \ldots, n\} \tag{10}$$

*and vectors $r, c, d \in \mathbb{R}^n$ which all sum to zero, such that*

$$C = B + r 1_n^T + 1_n c^T + diag(d). \tag{11}$$

*Moreover, this decomposition of $B$ is unique.*

Stated differently, the matrix $B$ will be a linear combination of matrices $r 1_n^T, 1_n c^T$ and $diag(d)$ which are in $\mathcal{V}_3, \mathcal{V}_4$ and $\mathcal{V}_2$, repsectively, and the matrix $C$ which is in $\hat{\mathcal{V}}$. Once we prove the lemma, we will conclude by showing that a matrix in $\hat{\mathcal{V}}$ can be written as a sum of matrices in $\mathcal{V}_5$ and $\mathcal{V}_6$, which will conclude the argument.

*Proof of the Lemma.* Let us denote the sum of the $i$-th row and column of $B$ by $r_i^B$ and $c_i^B$ respectively. Denote $d_i^B = B_{ii}$. Note that the vectors $r^B, c^B$, and $d^B$ all sum to zero by our assumption on $B$.

We need to show there exists vectors $r, d, c \in \mathbb{R}^n$ satisfying equation 11 for an appropriate $C \in \hat{\mathcal{V}}$. This will occur if and only if the following equations are satisfied

$$\sum_{i=1}^{n} r_i = 0$$

$$\sum_{i=1}^{n} c_i = 0$$

$$\sum_{i=1}^{n} d_i = 0$$

$$r_i + c_i + d_i = -d_i^B, \quad \forall i = 1, \ldots, n$$

$$n r_i + \sum_{j=1}^{n} c_j + d_i = -r_i^B, \quad \forall i = 1, \ldots, n$$

$$n c_i + \sum_{j=1}^{n} r_j + d_i = -c_i^B, \quad \forall i = 1, \ldots, n.$$

Here the first three constraints are the requirements that the vectors $r, c, d$ all sum to zero, and the last three constraints (each of which are actually $n$ constraints) follow from our requirement that the diagonal elements of $C$ will be zero and that the rows and columns of $C$ sum to zero.

Next, we note that since the vectors $r, c, d, r^B, c^B, d^B$ all sum to zero, if the last three equations above are satisfied for $i = 1, \ldots, n - 1$, they will automatically be satisfied for $i = n$. That is, these equations are equivalent to

$$\sum_{i=1}^{n} r_i = 0$$

$$\sum_{i=1}^{n} c_i = 0$$

$$\sum_{i=1}^{n} d_i = 0$$

$$r_i + c_i + d_i = -d_i^B, \quad \forall i = 1, \ldots, n - 1$$

$$n r_i + d_i = -r_i^B, \quad \forall i = 1, \ldots, n - 1$$

$$n c_i + d_i = -c_i^B, \quad \forall i = 1, \ldots, n - 1$$

This gives us a $3n$ linear equations system in $3n$ variables. We will show this equation has a unique solution and compute it explicitly. First, we eliminate the variables $r_n, c_n$ and $d_1, \ldots, d_n$ by setting

$$d_i = -d_i^B - r_i - c_i, \quad i = 1, \ldots, n - 1 \tag{12}$$

and

$$r_n = -\sum_{j=1}^{n-1} r_j, \quad c_n = -\sum_{j=1}^{n-1} c_j, \quad d_n = -\sum_{j=1}^{n-1} d_j \tag{13}$$

This ensures that the first four equations above are satisfied, and we are left with the task of choosing $r_1, \ldots, r_{n-1}, c_1, \ldots, c_{n-1}$ satisfying the last two equations (which are actually $2n - 2$ equations). By replacing the eliminated variables $d_i$ we obtain the equations

$$(n - 1) r_i - c_i = -r_i^B + d_i^B, \quad \forall i = 1, \ldots, n - 1$$

$$(n - 1) c_i - r_i = -c_i^B + d_i^B, \quad \forall i = 1, \ldots, n - 1$$

For every $i$, there are two equations in $r_i$ and $c_i$. Since we assume $n \geq 4$, this equation has a unique solution

$$\begin{pmatrix} r_i \\ c_i \end{pmatrix} = \begin{pmatrix} n-1 & -1 \\ -1 & n-1 \end{pmatrix}^{-1} \begin{pmatrix} -r_i^B + d_i^B, \\ -c_i^B + d_i^B, \end{pmatrix}$$

And we can derive

$$\begin{pmatrix} r_i \\ c_i \end{pmatrix} = \frac{1}{n^2 - 2n} \begin{pmatrix} n-1 & 1 \\ 1 & n-1 \end{pmatrix} \begin{pmatrix} -r_i^B + d_i^B, \\ -c_i^B + d_i^B, \end{pmatrix}$$

$$\begin{pmatrix} r_i \\ c_i \end{pmatrix} = \begin{pmatrix} nd_i^B - (n-1)r_i^B - c_i^B, \\ nd_i^B - (n-1)c_i^B - r_i^B, \end{pmatrix}$$

We can then reconstruct $d_1, \ldots, d_{n-1}$ from equation 12 and $c_n, r_n, d_n$ from equation 13. This concludes the proof. □

Given the proof, it remains to decompose a matrix $C \in \hat{\mathcal{V}}$ into a sum of two matrices in $\mathcal{V}_5, \mathcal{V}_6$. This can be simply done by writing

$$C = \frac{1}{2}(C + C^T) + \frac{1}{2}(C - C^T)$$

and noting that the matrices $\frac{1}{2}(C - C^T)$ and $\frac{1}{2}(C + C^T)$ are in $\mathcal{V}_5$ and $\mathcal{V}_6$.

## C.2 ISOMORPHISM TYPES

We now restate and prove the theorem on 2-IGN stated in the main text

**Theorem 4.1.** *For all $n \geq 4$, the space $\mathbb{R}^{n \times n}$ can be written as a direct sum of the spaces $\mathcal{V}_0, \ldots, \mathcal{V}_6$. These spaces are invariant and irreducible, and the isomorphism relations between them are given by $\mathcal{V}_0 \cong \mathcal{V}_1, \mathcal{V}_2 \cong \mathcal{V}_3 \cong \mathcal{V}_4$.*

*Proof.* The decomposition given in Subsection C.1 is unique when $n \geq 4$, and thus we see that

$$\mathbb{R}^{n^2} = \mathcal{V}_0 \oplus \ldots \oplus \mathcal{V}_6.$$

It is also clear that all the spaces $\mathcal{V}_i$ are invariant, that $\mathcal{V}_0, \mathcal{V}_1$ are isomorphic to the irreducible trivial space, and $\mathcal{V}_2, \mathcal{V}_3, \mathcal{V}_4$ are isomorphic to the irreducible space $\mathcal{V}(n)$. It remains to show: (a) That $\mathcal{V}_5, \mathcal{V}_6$ are not zero-dimensional, (b) that they are irreducible, and (c) that both $\mathcal{V}_5$ and $\mathcal{V}_6$ are not isomorphic to any of the other spaces.

We handle (a) by computing the dimension of $\mathcal{V}_5$ and $\mathcal{V}_6$. We note that a matrix $A \in \mathcal{V}_5$ can be parameterized by freely choosing all upper diagonal elements $A_{ij}$ with $j - i > 1$. The elements $A_{ji}$ with $j - i > 1$ are then determined by the constraint $A_{ji} = -A_{ij}$, and then the coordinates $A_{12} = -A_{21}, A_{23} = -A_{32}, \ldots$ are determined recursively by the constraint that the first $n - 1$ rows of $A$ will sum to zero. The obtained matrix $A$ satisfies $A = -A^T$ and its first $n - 1$ rows sum to zero. It follows from $A = -A^T$ that the sum of all elements of $A$ is zero and hence the last row of $A$ must also sum to zero and $A$ is indeed in $\mathcal{V}_5$.

To summarize, since $\mathcal{V}_5$ is parameterized by the upper diagonal elements $A_{ij}$ with $j - i > 1$ we deduce that

$$\dim(\mathcal{V}_5) = (n^2 - n)/2 - (n - 1) = \frac{n^2 - 3n}{2} + 1.$$

We compute the dimension of $\mathcal{V}_6$ by noting that $\hat{\mathcal{V}}$ from equation 10 is a direct sum of $\mathcal{V}_5$ and $\mathcal{V}_6$. The space $\hat{\mathcal{V}}$ can be parameterized by choosing an $(n - 1) \times (n - 1)$ matrix $A$ with zero on the diagonal, and whose elements sum to zero. There is then a unique extension of this matrix to an $n$ by $n$ matrix $\hat{A}$ which is in $\hat{\mathcal{V}}$ and satisfies $\hat{A}_{ij} = A_{ij}$ for all $1 \leq i, j \leq n - 1$. Accordingly, the dimension of $\hat{\mathcal{V}}$ is $(n - 1)^2 - (n - 1) - 1 = n^2 - 3n + 1$ and

$$\dim(\mathcal{V}_6) = \dim(\hat{\mathcal{V}}) - \dim(\mathcal{V}_5) = \frac{n^2 - 3n}{2}.$$

It follows that when $n \geq 4$, both $\mathcal{V}_5$ and $\mathcal{V}_6$ have positive dimension. Moreover, $\mathcal{V}_5$ and $\mathcal{V}_6$ are not isomorphic as they do not have the same dimension. The same argument shows that and $\mathcal{V}_6$ is are not isomorphic to the other $\mathcal{V}_i$ with $i = 0, \ldots, 4$ since they do not have the same dimension for any $n$. In contrast $\mathcal{V}_5$ does have the same dimension as $\mathcal{V}_2$ when $n = 4$, as we get $\dim(\mathcal{V}_5) = 3 = \dim(\mathcal{V}_2)$. Nonetheless, the spaces are still not isomorphic since the action of $S_n$ on the spaces is different. For example, the subspace of fixed points of the permutation $\tau = (1, 2)$ in the space $\mathcal{V}_2 \cong \mathcal{V}(n)$ is all $n = 4$ dimensional vectors $x$ which sum to zero and satisfy $x_1 = x_2$. This space is two dimensional. In contrast, one can verify that the subspace of fixed points of $\tau$ in $\mathcal{V}_5$ is one dimensional. Hence they cannot be isomorphic.

It remains to show that $\mathcal{V}_5$ and $\mathcal{V}_6$ are indeed irreducible. Let $\tilde{\mathcal{V}}_5 \subseteq \mathcal{V}_5$ and $\tilde{\mathcal{V}}_6 \subseteq \mathcal{V}_6$ be non-zero invariant subspaces. We need to show that $\tilde{\mathcal{V}}_j = \mathcal{V}_j$ for $j = 5, 6$. To show this, we consider the space
$$\mathcal{U} = \mathcal{V}_0 \oplus \mathcal{V}_1 \oplus \ldots \mathcal{V}_4 \oplus \tilde{\mathcal{V}}_5 \oplus \tilde{\mathcal{V}}_6.$$
It is sufficient to show that $\mathcal{U} = \mathbb{R}^{n \times n}$. Since $\mathcal{U}$ contains all diagonal matrices which are given by $\mathcal{V}_0 \oplus \mathcal{V}_2$, and since it is an invariant subspace, it is sufficient to show that the matrix $E^{1,2}$ which is zero in all coordinates except for a single coordinate $E^{1,2}_{1,2} = 1$, is in $\mathcal{U}$.

To show that $E^{1,2}$ is in $\mathcal{U}$, let us choose a non-zero matrix $A \in \tilde{\mathcal{V}}_5$ and $\mathcal{B} \in \tilde{\mathcal{V}}_6$. Both matrices have a non-zero off diagonal matrix element, and by applying an appropriate permutation and rescaling we can assume that $A_{12} = 1 = B_{12}$. Next, let us average over the group
$$stab(1, 2) = \{\tau \in S_n, \tau(1) = 1, \tau(2) = 2\}$$
to obtain new matrices
$$\hat{A} = \frac{1}{|stab(1,2)|} \sum_{\tau \in stab(1,2)} \tau \cdot A, \quad \hat{B} = \frac{1}{|stab(1,2)|} \sum_{\tau \in stab(1,2)} \tau \cdot B$$
Note that $\hat{A}$ (respectively $\hat{B}$) is in $\tilde{\mathcal{V}}_5$ (respectively $\tilde{\mathcal{V}}_6$) and satisfy $\hat{A}_{12} = 1 = \hat{B}_{12}$ and
$$\hat{A}_{1j} = \hat{A}_{1k}, \hat{B}_{1j} = \hat{B}_{1k} \forall j, k > 2$$
and
$$\hat{A}_{jk} = \hat{A}_{st}, \hat{B}_{jk} = \hat{B}_{st}$$
for all $j, k, s, t$ which are all larger than 2, and such that $j \neq k, s \neq t$. Defining for convenience
$$a = \frac{1}{n-2}, b = \frac{2}{(n-2)(n-3)},$$
it follows that
$$\hat{A} = \begin{pmatrix} 0 & 1 & -a & \ldots & -a \\ -1 & 0 & a & \ldots & a \\ a & -a & 0 & \ldots & 0 \\ \vdots & & & & \\ a & -a & 0 & \ldots & 0 \end{pmatrix} \quad \hat{B} = \begin{pmatrix} 0 & 1 & -a & \ldots & -a \\ 1 & 0 & -a & \ldots & -a \\ -a & -a & b & \ldots & b \\ \vdots & & & & \\ -a & -a & b & \ldots & b \end{pmatrix} - diag(0, 0, b, \ldots, b)$$
It is not difficult to show that any matrix which is constant along the rows is in $\mathcal{V}_0 \oplus \ldots \oplus \mathcal{V}_4 \subseteq \mathcal{U}$. The same is true for any matrix which is contant along the columns. Let $C(i)$ denote the matrix with $C_{ij}(i) = 1$ for all $j$ and $C_{kj}(i) = 0$ for all $j$ and all $k \neq i$. Then
$$\tilde{A} := \hat{A} + aC(1) - aC(2) - aC^T(1) + aC^T(2)$$
$$= \begin{pmatrix} 0 & 1+2a & 0 & \ldots & 0 \\ -1-2a & 0 & 0 & \ldots & 0 \\ 0 & 0 & 0 & \ldots & 0 \\ \vdots & & & & \\ 0 & 0 & 0 & \ldots & 0 \end{pmatrix}$$
$$\tilde{B} := \hat{B} + diag(0, 0, b, \ldots, b) - b1_{n \times n} + (a+b)(C(1) + C(2) + C^T(1) + C^T(2))$$
$$= \begin{pmatrix} 2a+b & 1+2a+b & 0 & \ldots & 0 \\ 1+2a+b & 2a+b & 0 & \ldots & 0 \\ 0 & 0 & 0 & \ldots & 0 \\ \vdots & & & & \\ 0 & 0 & 0 & \ldots & 0 \end{pmatrix}$$

where both $\tilde{A}$ and $\tilde{B}$ are in $\mathcal{U}$. Since $a, b$ are positive numbers, we can take the following linear combination which zeros out the $(2, 1)$ entry and sets the $(1, 2)$ entry to one:

$$\frac{1}{2}\left(\frac{1}{1+2a}\tilde{A} + \frac{1}{1+2a+b}\tilde{B}\right) = \begin{pmatrix} c & 1 & 0 & \ldots & 0 \\ 0 & d & 0 & \ldots & 0 \\ 0 & 0 & 0 & \ldots & 0 \\ \vdots & & & & \\ 0 & 0 & 0 & \ldots & 0 \end{pmatrix}$$

for appropriate $c, d$. Since diagonal matrices are in $\mathcal{U}$, we can remove the diagonal matrix $diag(c, d, 0, \ldots, 0)$ to finally obtain that the matrix $E^{1,2}$ is in $\mathcal{U}$. □

### C.3  SMALL $n$

As discussed in Finzi et al. (2021); Pearce-Crump (2022), when $n < 4$ the dimension of the space of equivariant mappings is smaller than $15$: it is $14$ for $n = 3$ and $8$ for $n = 2$. We now explain this according to our derivation.

For $n = 3$, the space $\mathcal{V}_6$ is zero-dimensional. In this case, there are only $2^2 + 3^2 + 1 = 14$ equivariant linear mappings. When $n = 2$, both $\mathcal{V}_5$ and $\mathcal{V}_6$ are zero dimensional. Additionally, we can directly verify that in this case, we can "drop" $\mathcal{V}_4$ as well because $\mathbb{R}^{2\times2} = \mathcal{V}_0 \oplus \mathcal{V}_1 \oplus \mathcal{V}_2 \oplus \mathcal{V}_3$. Since $\mathcal{V}_0 \cong \mathcal{V}_1$ and $\mathcal{V}_2 \cong \mathcal{V}_3$ we have $2^2 + 2^2 = 8$ equivariant linear mappings.

# D  PROOF FOR DEEP WEIGHT SPACE DECOMPOSITION

In this section we fill in some details on the tensor product of representations which were ommitted from the proof of Theorem 4.2 in the main text.

At the base of this discussion is the following lemma (Serre, 1977):

**Lemma D.1.** *Let $\mathcal{G}_1, \mathcal{G}_2$ be finite groups, and let $\mathcal{V}_1, \mathcal{V}_2$ be (real or complex) finite dimensional irreducible representations, with an action denoted by $\rho_1, \rho_2$. Then the action $\rho_1 \otimes \rho_2$ on $\mathcal{V}_1 \otimes \mathcal{V}_2$ is an irreducible representation of $\mathcal{G}_1 \times \mathcal{G}_2$.*

Firstly, we explain the decomposition of $\hat{\mathcal{W}}_m$ into irreducibles. We note that the representation $\hat{\mathcal{W}}_m$ of $\mathcal{G}$ can be thought of as a representation of $S_{d_{m-1}} \times S_{d_m}$, which is a tensor product of the representation $\mathbb{R}^{d_{m-1}}$ of $S_{d_{m-1}}$ and the representation $\mathbb{R}^{d_m}$ of $S_{d_m}$. Therefore (Serre, 1977), its irreducible decomposition is given by the tensor product of the irreducible decomposition of its factors, namely

$$\begin{aligned}
\hat{\mathcal{W}}_m &\cong \mathbb{R}^{d_{m-1}} \otimes \mathbb{R}^{d_m} \cong (\mathbf{S} \oplus \mathbf{V}(d_{m-1})) \otimes (\mathbf{S} \oplus \mathbf{V}(d_m)) \\
&\cong (\mathbf{S} \otimes \mathbf{S}) \oplus (\mathbf{S} \otimes \mathbf{V}(d_m)) \oplus (\mathbf{V}(d_{m-1}) \otimes \mathbf{S}) \oplus (\mathbf{V}(d_{m-1}) \otimes \mathbf{V}(d_m)) \\
&\cong \mathbf{S} \oplus \mathbf{V}(d_m) \oplus \mathbf{V}(d_{m-1}) \oplus \mathbf{M}_m.
\end{aligned}$$

All components in the decomposition are irreducible as a tensor product of irreducibles.

We also need to show that all irreducibles in the decomposition are absolutely irreducible. For $\mathbf{S}$ this is obvious. For all representations $\mathbf{V}(d_m))$ this follows from the fact that these representations can be identified with representations of the permutation group $S_{d_{m-1}}$, which are always absolutely irreducible. To see that $\mathbf{M}_m$ is absolutely irreducible, we note that $\mathbf{M}_m$ is the set of $d_{m-1} \times d_m$ real matrices whose rows and columns all sum to zero, and its complexification $c\mathbf{M}_m$ is the set of *complex* matrices whose rows and columns all sum to zero. $c\mathbf{M}_m$ is the tensor product of $c\mathbf{V}(d_{m-1})$ and $c\mathbf{V}(d_m)$, the complexification of $\mathbf{V}(d_{m-1})$ and $\mathbf{V}(d_m)$, respectively. Since these spaces are irreducible, so is their tensor product.

## D.1  DECOMPOSITION OF WEIGHT SPACES

We explain here how each matrix $A \in \mathcal{W}_m, m = 2, \dots, M-1$ can be written as a sum of elements from its irreducible decomposition. For convenience denote $a = d_m, b = d_{m-1}$. We want to write $A$ as a sum of elements from the spaces

$$\mathbf{S} \cong \{c1_{a \times b}\}, \quad \mathcal{V}_{m-1} \cong \{1_a c^T | c \in \mathbb{R}^b, c^T 1_b = 0\}, \quad \mathcal{V}_m \cong \{r1_b^T | r \in \mathbb{R}^a, r^T 1_a = 0\}$$

and

$$\mathbf{M}_m = \{C \in \mathbb{R}^{a \times b}, C1_b = 0_b, C^T 1_a = 0_a\}$$

As a first step, we take our given $A \in \mathcal{W}_m$ and write it as

$$A = \bar{A}1_{n \times n} + (A - \bar{A}1_{n \times n})$$

where $\bar{A}$ is the average of all elements of $A$, and $\bar{A}1_{n \times n}$ is in $\mathbf{S}$. Next, we need to decompose $(A - \bar{A}1_{n \times n})$ which we denote by $B$. Note that $\sum_{ij} B_{ij} = 0$.

Let $r$ and $c$ be the vectors describing the average of the rows and columns of $B$, respectively:

$$r = \frac{1}{b} B 1_b, \quad c = \frac{1}{a} B^T 1_a$$

Note that $r^T 1_a = \frac{1}{b} \sum_{ij} B_{ij} = 0$ and similarly $c^T 1_b = 0$ and thus $1_a c^T \in \mathcal{V}_{m-1}$ and $r1_b^T \in \mathcal{V}_m$. Now let us define

$$C = B - 1_a c^T - r1_b^T$$

We can directly verify that $C1_b = 0_b$ and $C^T 1_a = 0_a$. Therefore $C$ is in $\mathbf{M}_m$ and we obtained the decomposition we wanted.

# E   PROOFS FOR WREATH PRODUCTS

To conclude the proof of Theorem 5.1, we need to prove the following lemma

**Lemma E.1.** *Let $\mathcal{V}$ be an irreducible representation of $\mathcal{G}$, and assume the action of $\mathcal{G}$ on $\mathcal{V}$ is not trivial. Then $\mathcal{V}^k$ is an irreducible representation of $\mathcal{G} \wr S_k$.*

*Proof.* The space $\mathcal{V}^k$ is invariant. Let $\mathcal{W} \subseteq \mathcal{V}^k$ be a non-zero $\mathcal{G}^k$ invariant subspace. We need to show that $\mathcal{W} = \mathcal{V}^k$.

We first claim that there exists an element in $\mathcal{W}$ of the form $(u, 0, ..., 0)$ with $u \neq 0$. Let $(v_1, v_2, .., v_k)$ be a non-zero element in $\mathcal{W}$. By applying a permutation if necessary, we can assume without loss of generality that the first element is non-zero. If $v_i = 0, \forall 2 \leq i \leq k$, we are done. Otherwise, there exists some $g \in G$ such that $gv_1 \neq v_1$ and so

$$(g \cdot v_1, v_2, .., v_k) - (v_1, v_2, .., v_k) = (gv_1 - v_1, 0, .., 0) \in \mathcal{W}$$

The space $\{v \in \mathcal{V} | (v, 0, .., 0) \in \mathcal{W}\}$ is invariant under the action of $G$, and we just saw it is not zero. Since $\mathcal{V}$ is irreducible, this space equals to $\mathcal{V}$, and thus $\mathcal{W}$ contains $\mathcal{V} \oplus 0... \oplus 0$. By $S_k$ invariance it follows that $\mathcal{W}$ contains all $k$-tuples with a single non-zero entry, and since such tuples span all of $\mathcal{V}$ we deduce that $\mathcal{W} = \mathcal{V}$. □

**Theorem 5.2.** *Let $\mathcal{V}$ be a real representation of a finite group $\mathcal{G}$. and let $e_1, \ldots, e_s$ be a basis to the subspace $\mathcal{V}_{fixed}$. Let $\langle \cdot, \cdot \rangle$ be a $\mathcal{G}$ invariant inner product on $\mathcal{V}$. Then every linear equivariant map $L : \mathcal{V}^k \to \mathcal{V}^k$ is of the form*

$$L(v_1, \ldots, v_k) = \sum_{i,j=1}^{s} a_{ij} \left( \sum_{\ell=1}^{k} \langle v_\ell, e_i \rangle e_j, \ldots, \sum_{\ell=1}^{k} \langle v_\ell, e_i \rangle e_j \right) + \left( \hat{L}(v_1), \ldots, \hat{L}(v_k) \right) \quad (9)$$

*where $\hat{L} : \mathcal{V} \to \mathcal{V}$ is a linear equivariant map, and $a_{ij}$ are real numbers. Conversely, every linear mapping of the form defined in equation 9 is equivariant.*

*Proof.* It is straightforward to checking that every mapping of the form equation 9 is equivariant. We would like to prove the converse statement.

It is also straightforward to verify that the space spanned by the non-Siamese mappings

$$(v_1, \ldots, v_k) \mapsto (\sum_{\ell=1}^{k} \langle v_\ell, e_i \rangle e_j, \ldots, \sum_{\ell=1}^{k} \langle v_\ell, e_i \rangle e_j$$

does not depend on the choice of the basis $e_1, \ldots, e_s$ for $\mathcal{V}_{fixed}$. In particular, we can assume without loss of generality that $e_1, \ldots, e_s$ are an *orthonormal* basis for $\mathcal{V}_{fixed}$.

Note that $\mathcal{V}$ can be written as a direct sum of two $\mathcal{G}$ invariant spaces $\mathcal{V} = \mathcal{V}_{fixed} \oplus \mathcal{V}_\perp$, where $\mathcal{V}_\perp$ is the space of vectors in $\mathcal{V}$ which are orthogonal to $\mathcal{V}_{fixed}$ with respect to the given $\mathcal{G}$ invariant inner product. It follows that $\mathcal{V}^k = \mathcal{V}_{fixed}^k \oplus \mathcal{V}_\perp^k$ is a decomposition of $\mathcal{V}^k$ into two $\mathcal{G}^k$ invariant spaces.

In this situation, we know that any linear equivariant $L : \mathcal{V}^k \to \mathcal{V}^k$ can be written as $L = L_{fixed} + L_\perp$, where $L_{fixed} : \mathcal{V}_{fixed}^k \to \mathcal{V}_{fixed}^k$ and $L_\perp : \mathcal{V}_\perp^k \to \mathcal{V}_\perp^k$ are equivariant. We know that there are no linear equivariant maps from $\mathcal{V}_{fixed}$ to $\mathcal{V}_{perp}$ or vice versa, since $\mathcal{V}_{perp}$ cannot contain a trivial irreducible subspace.

Let $L : \mathcal{V}_\perp \to \mathcal{V}_\perp$ be a linear equivariant map. We want to show that it is a 'Siamese mapping'. Denote

$$L(v_1, \ldots, v_k) = (L_1(v_1, \ldots, v_k), \ldots, L_k(v_1, \ldots, v_k)),$$

We define $\hat{L} : \mathcal{V}_\perp \to \mathcal{V}_\perp$ by

$$\hat{L}(v) = L_1(v, 0, \ldots, 0).$$

Note that $\hat{L}$ is $\mathcal{G}$ equivariant because

$$\hat{L}(gv) = L_1(gv, 0, \ldots, 0) = gL_1(v, 0, \ldots, 0) = g\hat{L}(v), \quad \forall g \in \mathcal{G}.$$

Next, note that for every $v_2, \ldots, v_k \in \mathcal{V}_\perp$, we have that

$$L_1(0, v_2, \ldots, v_k) = L_1(g \cdot 0, v_2, \ldots, v_k) = g \cdot L_1(0, v_2, \ldots, v_k).$$

Thus, $L_1(0, v_2, \ldots, v_k) \in \mathcal{V}_\perp^k$, and since it is also in $\mathcal{V}_{fixed}^k$ we deduce that it is equal to zero. We deduce that

$$L_1(v_1, \ldots, v_k) = L_1(v_1, 0, \ldots, 0) = \hat{L}(v_1)$$

By $S_k$ equivariance we can deduce that

$$L_j(v_1, \ldots, v_j \ldots) = L_1(v_j, \ldots, v_1, \ldots) = \hat{L}(v_j)$$

and we can also show, as we did for $L_1$, that

$$L_j(v_1, \ldots, v_j \ldots) = L_j(0, \ldots, 0, v_j, 0 \ldots).$$

Thus $L$ is a Siamese mapping induced from $\hat{L}$, that is

$$L(v_1, \ldots, v_k) = \left( \hat{L}(v_1), \ldots, \hat{L}(v_k) \right)$$

Now, let us consider linear equivariant mappings from $\mathcal{V}_{fixed}^k$ to itself. Denote $\mathbf{S}_i = \{ce_i, c \in \mathbb{R}\}$, and note that an irreducible decomposition of $\mathcal{V}_{fixed}$ is given by

$$\mathcal{V}_{fixed} = \mathbf{S}_1 \oplus \ldots \oplus \mathbf{S}_s,$$

and $\mathcal{V}_{fixed}^k$ can be written as a direct sum of the invariant subspaces $\mathbf{S}_i^k$ for $i = 1, \ldots, s$. The action of $\mathcal{G}^k$ on $\mathbf{S}_i^k$ can be identified with the action of $S_k$ on these spaces, which is then isomorphic to the action of $S_k$ on $\mathbb{R}^k$, via the isomorphism $\psi_i : \mathbf{S}_i^k \to \mathbb{R}^k$ given by

$$\psi_i(v_1, \ldots, v_k) = (\langle v_1, e_i \rangle, \ldots, \langle v_1, e_k \rangle)$$

Note that since we assumed without loss of generality that $e_1, \ldots, e_s$ is an orthonormal basis, $\psi_i$ is zero on all other components of $\mathcal{V}^k$. Similarly we can define an isomrphism $\tilde{\psi}_i : \mathbb{R}^k \to \mathbf{S}_i^k$ by

$$\tilde{\psi}_i(x_1, \ldots, x_k) = (x_1 e_i, \ldots, x_k e_i).$$

As discussed in Section 4.1, an equivariant mapping from $\mathbb{R}^k$ to itself can be written as a linear combination of the identity mapping $x \mapsto x$ (which is a Siamese mapping), and the mapping

$$Tx = \left( \sum_{i=1}^{n} x_i, \ldots, \sum_{i=1}^{n} x_i \right)$$

Accordingly, a mapping from $\mathbf{S}_i^k$ to $\mathbf{S}_j^k$ is a linear combination of a Siamese map and the mapping $\tilde{\psi}_j \circ T \circ \psi_i$ which is given by

$$\tilde{\psi}_j \circ T \circ \psi_i(v_1, \ldots, v_k) = \left( \sum_{\ell=1}^{k} \langle v_\ell, e_i \rangle e_j, \ldots, \sum_{\ell=1}^{k} \langle v_\ell, e_i \rangle e_j \right).$$

This concludes the proof.

$\square$

### E.1 BEYOND $S_k$

We now generalize the results from Theorem 5.2 to the case where $S_k$ is replaced with any subgroup $H \leq S_k$ which acts on $\{1, \ldots, k\}$ transitively:

**Theorem E.2** (Generalization of Theorem 5.2). *Let $\mathcal{V}$ be a real representation of a finite group $\mathcal{G}$. and let $e_1, \ldots, e_s$ be a basis to the subspace $\mathcal{V}_{fixed}$. Let $\langle \cdot, \cdot \rangle$ be a $\mathcal{G}$ invariant inner product on $\mathcal{V}$.*

*Let $H$ be a subgroup of $S_k$ which acts on $\{1, \ldots, k\}$ transitively. Let $T_1, T_2, \ldots, T_h$ be a basis for the $H$-equivariant mappings from $\mathbb{R}^k$ to itself, where $T_h$ is the identity mapping.*

*Then every $\mathcal{G} \wr H$ linear equivariant map $L : \mathcal{V}^k \to \mathcal{V}^k$ is of the form*

$$L(v_1, \ldots, v_k) = \sum_{\ell=1}^{h-1} \sum_{i,j=1}^{s} a_{ij\ell} \cdot (\tilde{\psi}_j \circ T_\ell \circ \psi_i)(v_1, \ldots, v_k)$$
$$+ \left( \hat{L}(v_1), \ldots, \hat{L}(v_k) \right)$$

*where $\hat{L} : \mathcal{V} \to \mathcal{V}$ is a linear equivariant map, and $a_{ij\ell}$ are real numbers. Conversely, every linear mapping of the form defined in equation 9 is equivariant.*

The proof is identical to the proof of Theorem 5.2.

