# OpenReview forum: "REVISITING MULTI-PERMUTATION EQUIVARIANCE THROUGH THE LENS OF IRREDUCIBLE REPRESENTATIONS"
_ICLR.cc/2025/Conference — ICLR 2025 Poster_

### Official Review · Reviewer_QTpm · 2024-10-17

**Soundness:** 3
**Presentation:** 4
**Contribution:** 2
**Rating:** 6
**Confidence:** 4

**Summary:**

The paper considers the problem of constructing linear equivariant layers for groups acting (linearly) on input and output spaces. Specifically, it proposes to exploit the decomposition into irreducible group representations and then appealing to Schur’s Lemma, which reduces the problem to choosing coefficients for pairs of isomorphic representations. Several specific instances are analyzed, such as permutation groups in the context of graph neural networks, groups acting on weights of deep networks, and wreath products acting on products of representations.

**Strengths:**

-The paper is exceptionally well written. The language is clear and concise, the sections are structured, and the mathematical formalism/notation is elegant.

-The problem considered is a fundamental one in machine learning literature. Constructing (linear) equivariant maps lies at the heart of geometric deep learning, which has been successful in several applications.

-The proposed solution is general, as it applies, in principle, to any input/output group representation. Several existing frameworks are phrased under the same paradigm, contributing with structure and clarity to the geometric deep learning literature.

**Weaknesses:**

I believe that the proposed approach via Schur’s Lemma comes with disadvantages. To begin with, using Schur’s Lemma to construct equivariant linear maps is not novel in the geometric deep learning community. It is a rather well-known technique – see, for example, Behboodi et al., Section 3.2. This is a major concern, since Schur’s Lemma represents a core point of this work; the other contributions amount to rephrasings of known frameworks from the literature under the lenses of Schur’s Lemma. Moreover, Schur’s Lemma has some restrictions. First, it requires the decomposition into irreducible representations to be known a priori, which is not always the case. Such decomposition is challenging to compute algorithmically for general groups and representations.  Second, Schur’s Lemma applies naively only to complex representations (i.e., over $\mathbb{C}$). As the authors mention, this is not an issue for permutation groups (appendix B), but it can be for other groups. It is still possible to apply Schur’s Lemma to arbitrary real representations of arbitrary groups, but this involves subtleties – see Behboodi et al., Section 8.

I also find the experimental section rather weak. The experiments reported only consider ideal equivariant tasks, i.e., scenarios where the ground-truth function is equivariant. The experimental results show that adding equivariant layers to the network improves (generalization) performance, as compared to non-equivariant architectures. This is not surprising, since in these cases the inductive bias given by equivariance aligns perfectly with the structure of the task. In typical real-world scenarios (e.g., image classification), the (highly-noisy) ground-truth function is instead not exactly equivariant, or it is not equivariant on all the input data. In my opinion, it would be more informative and less trivial to test the models on these types of real-world tasks. The equivariance bias is often still beneficial in terms of generalization – as works in geometric deep learning have extensively shown – but empirical investigations are required to assess this carefully.

Minor typos:

-The paragraph title on line 86 is not capitalized, while the one on line 100 is.

-The tables in section 6 exceed the margins of the paper.


Behboodi et al., A PAC-Bayesian Generalization Bound for Equivariant Networks, NeurIPS 2022.

**Questions:**

I would like the authors to comment on the above points regarding novelty and significance of experiments.

My current opinion is that the work is exceptionally well-written, and bears several contributions to the geometric deep learning literature. However, I am concerned with the novelty and significance, as outlined above. Still, I am leaning towards accepting the paper, but would like to hear from the authors about my points of criticism.

---

> ### Author Response · Authors · 2024-11-21
> **Answer to Reviewer**
>
> We thank the reviewer for their thoughtful comments and time.
>
> **W1:** *I believe that the proposed approach via Schur’s Lemma comes with disadvantages. To begin with, using Schur’s Lemma to construct equivariant linear maps is not novel in the geometric deep learning community. It is a rather well-known technique – see, for example, Behboodi et al., Section 3.2.*
>
> **A1:**
> We do not dispute this claim. Indeed, in the related work section, we have devoted a paragraph to reviewing works that have employed this approach. The point of this paper is to apply the irreducible representation methodology to important equivariant learning scenarios where it has not yet been applied and show that this approach yields some benefits over the more commonly used parameter-sharing analysis.
>
> **W2:**
> *“This is a major concern, since Schur’s Lemma represents a core point of this work; the other contributions amount to rephrasings of known frameworks from the literature under the lenses of Schur’s Lemma.”*
>
> **A2:** We disagree with this claim. Section 5 offers a novel characterization of wreath-equivariant linear layers, a key result. Unlike [Wang et al., Neurips 2020], which focused only on transitive permutation actions with few non-siamese layers, we provide a complete, general characterization, showing many non-siamese layers exist. Experiments confirm their relevance for learning Wasserstein distances and aligning deep weight spaces.
> The first part of the paper also adds value by simplifying the derivation of DWS layers, previously tedious. This clarity benefits researchers improving architectures in this emerging field.
>
> **W3:**
> *Moreover, Schur’s Lemma has some restrictions. First, it requires the decomposition into irreducible representations to be known a priori, which is not always the case. Such decomposition is challenging to compute algorithmically for general groups and representations. Second, Schur’s Lemma applies naively only to complex representations . As the authors mention, this is not an issue for permutation groups (appendix B), but it can be for other groups. It is still possible to apply Schur’s Lemma to arbitrary real representations of arbitrary groups, but this involves subtleties – see Behboodi et al., Section 8.*
>
> **A3:**
> We agree with all the facts stated in this paragraph. Using irreducible representations to characterize linear layers has advantages and disadvantages. You summarized them well. The point of this manuscript is to show the benefits of this approach for some permutation actions where this approach is not typically used. In particular, (1) we get much simpler derivations of the DWS layers, and (2) we derive all wreath equivariant layers, which was not done previously.
>
> **Q4:**
> *“I also find the experimental section rather weak. The experiments reported only consider ideal equivariant tasks, i.e., scenarios where the ground-truth function is equivariant. The experimental results show that adding equivariant layers to the network improves (generalization) performance as compared to non-equivariant architectures. This is not surprising since in these cases, the inductive bias given by equivariance aligns perfectly with the structure of the task. In typical real-world scenarios (e.g., image classification), the (highly noisy) ground-truth function is instead not exactly equivariant, or it is not equivariant on all the input data. In my opinion, it would be more informative and less trivial to test the models on these types of real-world tasks. The equivariance bias is often still beneficial in terms of generalization – as works in geometric deep learning have extensively shown – but empirical investigations are required to assess this carefully.”*
>
> **A4:**
> It’s possible that we didn’t explain ourselves well, but the baselines are some Siamese models that are also equivariant to wreath products. The difference is that our method includes all equivariant layers and, hence, is more expressive and leads to better results in practice.
>
> Regarding equivariance: in our view, equivariant is often exact and not approximate, even with image classification: e.g., given a noisy image of a motorbike, a rotated image of the image will stil be an image of a motorbike.
>
> Regarding “real world tasks”: we consider the results in Table 2 and Table 3 “real world tasks”. They address the problem of computing Wasserstein distances, and aligning neural weights spaces, which are important and relevant topics. The datasets and competing siamese methods come from recent Neurips/ICML submissions [Chen and Wang, Neurips 2023] and [Navon et al. ICML 2024]. They involve computing Wasserstein distances on 3D models from ModelNet40 and for RNA benchmarks and aligning neural networks trained on MNIST and CIFAR.
>
>  **Minor Typos:**
> *The paragraph title on line 86 is not capitalized, while the one on line 100 is.*
> *The tables in section 6 exceed the margins of the paper.*
>
> **A** fixed

---

> > ### Comment · Reviewer_QTpm · 2024-11-23
> >
> > I am grateful to the authors for their reply.
> >
> > I acknowledge that the contribution on wreath-equivariant layers is worthy of interest, and I will therefore keep my (positive) score.

---

### Official Review · Reviewer_mzEE · 2024-11-02

**Soundness:** 4
**Presentation:** 3
**Contribution:** 2
**Rating:** 6
**Confidence:** 3

**Summary:**

The paper introduces a novel methodology for characterizing equivariant linear layers for permutation representations, utilizing classical results from representation theory. Specifically, it provides an alternative characterization of equivariant linear layers for DeepSets, $2$-IGNs, and DWSNets, as well as the first comprehensive characterization of equivariant linear layers for unaligned symmetric elements. Importantly, the authors identify novel non-Siamese layers and empirically assess their impact.

**Strengths:**

- Clear presentation and notation, supported by rigorous proofs.
- The methodology is both valuable and simple, with potential to generalize beyond the examples presented.
- A novel and complete characterization of representations for unaligned symmetric elements.

**Weaknesses:**

- Lacks discussion on extending the approach to groups and representations beyond the few presented.
- In particular, an appropriate discussion on characterizing the more expressive layers of $k$-IGNs for $k>2$ is missing.

**Questions:**

1. I find the methodology presented in L135-155 valuable to the research community due to its generalizability beyond the provided examples, most of which are already characterized. For this reason, would it be possible to add a *brief* discussion on the generalization of this methodology to strengthen the impact of this contribution and broaden its relevance to a wider community? See the following for more specific questions.
2. Computing a basis compatible with the irreducible representation decomposition can be challenging. Does this difficulty limit the methodology’s generalization? Are there similar technical challenges for characterizing $k$-IGN layers for $k > 2$?
3. Can this methodology be applied to other groups beyond $S_n$ and wreath products? If so, could you briefly provide a few examples?
4. Representations of the symmetric group are relevant in machine learning and its irreducible representation are absolutely irreducible. In contrast, other relevant groups, such as finite cyclic groups, have real irreducible representations that are not absolutely irreducible. Could the framework presented here be extend to these cases? What potential challenges do you envision in extending to non-absolutely irreducible representations?
5. Could you elaborate on the future directions for $k$-IGNs presented in the conclusions (L537-539)?

**Minor Issues (No Impact on Recommendation):**
- L073: I recommend specifying "$2$-IGNs" for transparency.
- L183: Is the presentation of $P_\tau$ unnecessary?
- L340: The wreath product of groups is introduced but not defined in detail; as this operation is uncommon in machine learning literature, additional explanation would benefit Section 5. Also, consider demonstrating that equation 7 forms a linear representation of this group, perhaps in the appendix.
- L420: Typo, “is prove”.
- L379 and L1030: I cannot understand why $\mathcal{V}^k$ is an irreducible representation of $\mathcal{G}^k$; is it instead irreducible for $\mathcal{G} \wr S_n$?
- L1040: The closing curly bracket is missing.

---

> ### Author Response · Authors · 2024-11-21
> **Answer to Reviewer**
>
> We thank the reviewer for their thoughtful comments and time.
>
> **Weaknesses**:
> “In particular, an appropriate discussion on characterizing the more expressive layers of k-IGNs for is missing”.
>
> **Answer**
> Regarding k-IGN for k>2, we currently do not know how to compute the irreducible decomposition in this scenario.
>
> **Questions**
>
> **Q1:** For this reason, would it be possible to add a brief discussion on the generalization of this methodology to strengthen the impact of this contribution and broaden its relevance to a wider community? See the following for more specific questions”.
>
> **A1:**
> Thanks for this great suggestion. We added the following discussion based on your questions to page 3: *“We note that the cornerstones of this methodology:  decomposition into irreducibles and Schur's Lemma, are applicable for all finite dimensional representations of finite groups (and also for compact infinite groups like $SO(d)$). The main challenge in this approach is characterizing and computing the decomposition into irreducibles. This needs to be done on a case to case basis. Much of the remainder of the paper will be devoted to computing these decompositions for important equivariant learning scenarios.”*
>
> **Q2**: Does this difficulty limit the methodology’s generalization? Are there similar technical challenges for characterizing k-IGN layers for k>2?”
>
> **A2:**
> Yes, in this approach the main challenge is characterizing the irreducible representations. Indeed this is what stops us from applying it to k-IGN for k>2. We know these irreducible exists, but we are still missing an algorithm to compute the decomposition.  Note that, although characterization for k-IGN is difficult, characterization for DWS layers is very simple in our methodology in contrast to other methods that are very tedious and difficult to understand.
>
> **Q3:** Can this methodology be applied to other groups beyond S_N and wreath products? If so, could you briefly provide a few examples?
>
> **A3:**
> Generally, the irreducible-based method can be applied to all finite groups that are absolutely irreducible (see discussion in next question). It can also be applied to infinite compact groups like SO(3), which is actually a popular approach. Examples of papers following this approach are given in the related work section.
>
> **Q4:** Could the framework presented here be extend to these cases? What potential challenges do you envision in extending to non-absolutely irreducible representations?
>
> **A4:**
> For general real representations, we can still write the representation as a sum of irreducible representations, and there will still be no linear equivariant maps between non-isomorphic irreducibles. The difference is that the space of linear equivariant maps from an irreducible V to itself will be either
> one dimensional, {$\lambda I| \lambda \in R$} as in the permutation case
> two-dimensional, isomorphic to the complex numbers
> four-dimensional, isomorphic to the quaternions.
> This is explained nicely here https://math.mit.edu/~poonen/715/real_representations.pdf
> One would then apriori need to check case by case what the space of isomorphisms is for each irreducible. An interesting alternative would be to use an automatic numerical method to find all equivariant layers between the irreducibles, as described in [Finzi et al. 2021]. Since the dimensions of the equivariant layers is at most four, the computational price of such an approach should be very reasonable.
>
> We have added a discussion of this point to page 3 as well:
> “We note that when V is not absolutely irreducible,
> the space of isomorphisms from V to W is either 2 or 4-dimensional (Poonen, 2016). In this setting
> , using an automatic computational method to find all equivariant layers may be beneficial (Finzi et al. 2021).”
>
> **Q5:** Could you elaborate on the future directions for-IGNs presented in the conclusions (L537-539)?”
>
> **A5:**
> Yes. The standard k-IGN framework consider equivariant maps between tensor representations, which are of dimension $n,n^2,n^3,...$ The irreducible representation framework shows that, e.g. the $n^2$ dimensional matrix space from 2-IGN can be decomposed into 7 irreducible subspaces: two are 1 dimensional, three are $n-1$ dimensional, and the remaining two are approximately $n^2/2$ dimensional. One could then consider equivariant maps based on this decomposition, with a different number of hidden features coming from each one of the irreducible representations. e.g., from a computational perspective, it may make sense to take more features from the $n-1$ dimensional representations and fewer features from the $n^2/2$ dimensional representations. Similar ideas could be applied to k-IGN (but this would first require characterizing the irreducibles which is also future work).
>
> **Minor Issues (No Impact on Recommendation)...**
>
> Thanks, we incorporated all your suggestions in the revised manuscript.

---

> > ### Comment · Reviewer_mzEE · 2024-11-24
> >
> > I thank the authors for their responses and maintain my positive assessment and score.

---

### Official Review · Reviewer_vHFp · 2024-11-03

**Soundness:** 3
**Presentation:** 3
**Contribution:** 3
**Rating:** 6
**Confidence:** 4

**Summary:**

The paper studies equivariant linear layers for representations of permutations and related groups from a novel irreducible representations perspective. The authors provide an alternative derivation for models including DeepSets, 2-IGN, and Deep Weight Space (DWS) networks. The theory is then extended to unaligned symmetric sets, showing that there is a vast number of additional non-Siamese layers in certain settings. Experiments show that additional non-Siamese layers improve the performance in tasks like graph anomaly detection, weight space alignment, and learning Wasserstein distances.

**Strengths:**

The paper offers the irreducible representations perspective for deriving classical models like DeepSets, 2-IGN and DWS networks. Some derivations are simpler than the original ones. The writing is clear and easy to follow. I check with the details and they are sound.

**Weaknesses:**

* While the new derivations align with original methods, the resulting models are not new. The concept of ``irreducible representation'' is also well studied, so the contribution of the paper lies mainly in bridging two topics, which is interesting but natural. In particular for equivariant graph layers, the authors only provide derivations for 2-IGN. As admitted in the limitation section, the paper does not involve higher-order $k$-IGN. The author should explain whether their method is broadly applicable for these networks based on tensor representations, or need case-by-case derivations.

* Although this is a theoretical paper, the experiments could be improved. More baselines and more real-world tasks are strongly encouraged.

**Questions:**

* Can the method be generalized to higher-order $k$-IGN in a principled manner? Can you briefly describe the claim that ``using irreducibles could lead to new equivariant models with intermediate irreducible features of lower dimensions''?

* Can you conduct more experiments on real-world and large-scale datasets, and include more baseline? In addition, can you intuitively explain why non-Siamese layers help in these tasks?

---

> ### Author Response · Authors · 2024-11-21
> **Answer to Reviewer**
>
> We thank the reviewer for their time and thoughtful comments.
>
> **Weaknesses**
>
> **W1:**
> While the new derivations align with original methods, the resulting models are not new. The concept of ``irreducible representation'' is also well studied, so the contribution of the paper lies mainly in bridging two topics, which is interesting but natural.
>
> **A1:**
> We do not agree with this description. Firstly, our results and model for sets of unaligned symmetric elements (Section 5)  are completely new.  Secondly, also the theoretical results which are a new derivation of existing models require a non-trivial analysis of the problem and are not all immediate. The generally known fact is that irreducible representations exist and that if the decomposition to irreducibles is known, it can be used to characterize all linear equivariant layers via Schur’s lemma. However, the decomposition into irreducibles for the examples we discussed was unknown.
>
> **W2:** In particular for equivariant graph layers, the authors only provide derivations for 2-IGN. As admitted in the limitation section, the paper does not involve higher-order-IGN. The author should explain whether their method is broadly applicable for these networks based on tensor representations, or need case-by-case derivations”
>
> **A2:**
> Currently, our analysis does not support higher order IGNs. We know theoretically that the tensor representations can be decomposed into irreducibles, but we currently do not know how to compute this decomposition. We have not invested much energy into this question because the characterization of k-IGN layers using parameter sharing is very elegant. In contrast, the advantage of our approach is more apparent for Deep Weight Spaces or Wreath equivariant structures.
>
> **W3:** Although this is a theoretical paper, the experiments could be improved. More baselines and more real-world tasks are strongly encouraged
>
> **A3:** We include three experiments in the paper: one indeed synthetic experiment on graph anomaly detection, and two experiments improving upon recent successful Siamese-based methods for (a)  computing Wasserstein distance (Chen and Wang, Neurips 2023) and (b) aligning weight spaces (Navon et al, ICML 2024). We believe the experimental part is on par, or more extensive than, what is common in similar ICLR theoretical papers.
>
> **Questions:**
>
> **Q1:** Can the method be generalized to higher-order-IGN in a principled manner?
>
> **A1:**
>  Currently, our work doesn’t generalize to higher order-IGN (discussed more above in answer to **W2**)
>
> **Q2:**
> *Can you briefly describe the claim that ``using irreducibles could lead to new equivariant models with intermediate irreducible features of lower dimensions''?”*
>
> **A2:** The standard k-IGN framework consider equivariant maps between tensor representations, which are of dimension $n,n^2,n^3,...$ The irreducible representation framework shows that e.g., the $n^2$ dimensional matrix space from 2-IGN can be decomposed into 7 irreducible subspaces: two are 1 dimensional, three are n-1 dimensional, and the remaining two are approximately $n^2/2$ dimensional. One could then consider equivariant maps based on this decomposition, with a different number of hidden features coming from each one of the irreducible representations. e.g., from a computational perspective, it may make sense to take more features from the $n-1$ dimensional representations and fewer features from the $n^2/2$ dimensional representations. Similar ideas could be used for general k-IGN (once the decomposition is computed)
>
> **Q3:** Can you conduct more experiments on real-world and large-scale datasets, and include more baseline? In addition, can you intuitively explain why non-Siamese layers help in these tasks?
>
> **A3:**
> We were not able to add more tasks in the time allotted for the rebuttal. We think that the current scope of experiments is on par, or more extensive than, what is common in similar ICLR theoretical papers.

---

> > ### Comment · Reviewer_vHFp · 2024-11-25
> >
> > I thank the authors for their response. Although there are still some unaddressed concerns, I agree that the paper has many contributions and thus will keep my rating.

---

### Official Review · Reviewer_iEGE · 2024-11-04

**Soundness:** 2
**Presentation:** 2
**Contribution:** 2
**Rating:** 3
**Confidence:** 3

**Summary:**

The paper introduces an alternative approach for characterizing equivariant linear layers in neural networks that process permutation and related group representations. The paper derives a simpler method for obtaining existing models such as DeepSets, 2-IGN, and Deep Weight Space networks, based on irreducible representations and Schur’s lemma. The proposed framework also considers unaligned symmetric sets, that build upon equivariance to the wreath product of groups.

**Strengths:**

1. The paper introduces a fresh perspective on equivariant layer characterization by applying irreducible representations and Schur’s lemma to obtain simplified derivations of established models, such as DeepSets, 2-IGN, and Deep Weight Space (DWS) networks.

2. The theoretical foundations are well-developed. The work provides a complete characterization of equivariant layers in the context of unaligned symmetric sets, which is an interesting theoretical contribution.

**Weaknesses:**

1. The presentation and flow of the paper could be improved. The claims and results are challenging to follow, which may limit the broader audience’s ability to appreciate the work.

2. The paper’s contributions lack clarity. The paper offers an irreducible-based derivation for existing results and characterizes equivariant functions on unaligned symmetric elements, but the impact and relevance of these contributions remain unclear. It is not evident how these results benefit the design of novel architectures or enhance our understanding of current ones. This limits the significance of the work and may fall short of ICLR’s standards.

3. The empirical evaluation is limited, and the results are not compelling. Using synthetic data for anomaly detection does not sufficiently demonstrate the method’s practical applicability, as the task is relatively unchallenging and does not show the strengths of the proposed approach.

**Questions:**

Please see Weaknesses.

---

> ### Author Response · Authors · 2024-11-21
> **Answer to review**
>
> We thank the reviewer for their thoughtful comments and time.
>
> **W1:**
>
>  “The presentation and flow of the paper could be improved. The claims and results are challenging to follow, which may limit the broader audience’s ability to appreciate the work.”
>
> **A1:**
> Note that all other reviewers were very positive about the paper's clarity. If there are any specific points you think we should clarify, we will be happy to do so.
>
> **W2:**
> The paper’s contributions lack clarity. The paper offers an irreducible-based derivation for existing results and characterizes equivariant functions on unaligned symmetric elements, but the impact and relevance of these contributions remain unclear. It is not evident how these results benefit the design of novel architectures or enhance our understanding of current ones. This limits the significance of the work and may fall short of ICLR’s standards.
>
> **A2:**
> Our theoretical contribution is divided into two parts: revisiting existing results and new results for sets of unaligned symmetric elements. In particular, the characterization of DWS layers is very challenging to understand using other methods and requires many bookkeeping and scenario splitting. Our DWS derivation is very simple to understand. We believe this will be helpful for researchers working on improving architectures in this emerging topic.
>
> The second result for sets of unaligned symmetric elements is completely new and arises in many real-world scenarios, including graph anomaly detection, Learning Wasserstein distance and Weight space alignment problems discussed in the paper. Other examples not discussed in the paper include learning ICP-like metrics [1] or graph matching [2]. The main insight in this paper, for these problems, is that besides the commonly used siamese structrure, there can be a considerable number of  non-siamese layers which respect the problem’s equivariant structure, and characterizing all these layers.
>
> **W3:**
> The empirical evaluation is limited, and the results are not compelling. Using synthetic data for anomaly detection does not sufficiently demonstrate the method’s practical applicability, as the task is relatively unchallenging and does not show the strengths of the proposed approach.”
>
> **A3:**
> While the anomaly detection experiment is indeed synthetic and highlights our theoretical advantage, we do empirically evaluate our approach on two additional real-world datasets that were used in other ICML, NIPS conferences. In the Wasserstein distance approximation experiment, we compare against a recent Siamese method from NeurIPS 2023, on a variety of datasets including a gene expression dataset (RNAseq) and an object point-cloud dataset ModelNet40. In the weight space alignment we compare against a Siamese method from  ICML 2024,  testing our performance on implicit neural representations (INRs) of MNIST and CIFAR10 image datasets.
>
>
> [1] Deep Closest Point: Learning Representations for Point Cloud Registration, Wang and Solomon, ICCV 2019
>
> [2] Neural Graph Matching Network: Learning Lawler’s Quadratic Assignment Problem With Extension to Hypergraph and Multiple-Graph Matching
> Wang, Yan and Yang, TPAMI 2022

---

> > ### Author Response · Authors · 2024-11-27
> >
> > Dear reviewer, have our answers addressed your concerns? We're looking forward to  your feedback.

---

### Author Response · Authors · 2024-11-21
**For all reviewers**

We’d like to thank the reviewers for taking the time to review the paper and their constructive comment. We were happy to see that for the most part the reviewers were positive about the paper, and felt that it was “exceptionally well-written, and bears several contributions to the geometric deep learning literature”.

There were several points that came up in several reviews, which we would like to clarify:
* Some of the reviewers claimed that  “the contributions amount to rephrasings of known frameworks from the literature under the lenses of Schur’s Lemma.” We’d like to emphasize that this is not the case. Section 5 provides a characterization of wreath-equivariant linear layers which is completely new. This is, in our view, an important result. Previous work [Wang et al., Neurips 2020] devoted solely to this problem have only focused on the special case where the permutation action is transitive, and suggested a very small number of non-siamese layers. We give a complete characterization in a much more general setting, and show that in several cases there is a vast number of non-siamese layers. We also experimentally show the relevance of these results for learning Wasserstein distances  and aligning deep weights spaces.
We also feel the contribution in Section 4, which is indeed dedicated to derivation of known results from the irreducible perspective, will also be  valuable to the community. In particular, our derivation of Deep Weight Space layers is substantially simpler than previous methods, and we believe this will be helpful for researchers working on improving architectures in this emerging topic.

* Several reviewers remarked on the fact that we do not include a derivation of k-IGN for k>2. Indeed we currently cannot do this. We have also not invested much energy into this question because the characterization of k-IGN layers using parameter sharing is very elegant. In contrast, the advantage of our approach is more apparent for Deep Weight Spaces or Wreath equivariant structures.

* There were some remarks on the lack of “real world experiments”. In this context we’d like to emphasize that we consider the results in Table 2 and Table 3 “real world tasks”. They address the problem of computing Wasserstein distances, and aligning neural weights spaces, which are important and relevant topics, and the datasets and competing siamese methods come from recent successful submissions [Chen and Wang, Neurips 2023] and [Navon et al. ICML 2024]. We believe the magnitude of the  experimental section is adequate for a theory based paper like ours.

We have uploaded a revised version of our manuscript, addressing your comments. Changes from the submitted version are marked in blue.

---

### Meta-Review · Area_Chair_D3Vs · 2024-12-18

**Metareview:**

This paper derives the existing equivariant models of Deep Sets, 2-IGNs, and Deep Weight Space networks, in terms of irreducible representations and Schur's lemma. The concept of the paper is interesting, and the theoretical contribution delivers a mathematical approach to understanding existing machine learning architectures that a priori could seem ad hoc. However, the reviewers raised valid concerns regarding the applicability of the approach in general (Schur's lemma holds over $\mathbb C$, and the approach is only described for permutation groups for which the irreps are simple to compute). They also raised concerns about the experimental evaluation. However, using standard mathematical tools to explain things that could seem an otherwise arbitrary construction is useful, so the positives outweigh the negatives.

**Additional Comments On Reviewer Discussion:**

All but one reviewer found the paper marginally above the threshold after the discussion period. The most negative reviewer voted to reject (3) but did not engage in the discussion, so after a conversation with the senior area chair, we decided to discard this review.

---

### Decision · Program_Chairs · 2025-01-22

Accept (Poster)